# On-shell effective theory for higher-spin dark matter

Adam Falkowski[1], Giulia Isabella[1] and Camila S. Machado[2]

**1** Université Paris-Saclay, CNRS/IN2P3, IJCLab, 91405 Orsay, France
**2** Deutsches Elektronen-Synchrotron (DESY), D-22607 Hamburg, Germany

## Abstract

We apply the on-shell amplitude techniques in the domain of dark matter. Without evoking fields and Lagrangians, an effective theory for a massive spin-*S* particle is defined in terms of on-shell amplitudes, which are written down using the massive spinor formalism. This procedure greatly simplifies the study of theories with a higher-spin dark matter particle. In particular, it provides an efficient way to calculate the rates of processes controlling dark matter production, and offers better physical insight into how different processes depend on the relevant scales in the theory. We demonstrate the applicability of these methods by exploring two scenarios where higher-spin DM is produced via the freeze-in mechanism. One scenario is minimal, involving only universal gravitational interactions, and is compatible with dark matter masses in a very broad range from sub-TeV to the GUT scale. The other scenario involves direct coupling of higher-spin DM to the Standard Model via the Higgs intermediary, and leads to a rich phenomenology, including dark matter decay signatures.

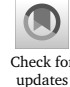

## 1  Introduction

On-shell amplitude methods, initially developed in the context of massless QCD [1], are finding applications in more and more areas of particle physics. Recently these methods have shed some new light on dynamics of black holes [2–5], construction of bases of effective field theories (EFTs) [6–10], calculation of renormalization group running [11–14], to name just a few examples. An important ingredient in this program was the development of a convenient spinor formalism to handle massive particles [15]. This opened the opportunity for new applications within broad classes of theories, such as spontaneously broken gauge theories, massive gravity, and models with higher-spin particles. In this paper we take the on-shell amplitude methods to the field of dark matter (DM).

In the past decades, an ever-growing number of experiments has enabled us to target a large number of DM models and cover extensive regions of their parameter space. In spite of this rich program, the nature of DM remains elusive, and it is conceivable that our model-building efforts so far have been misdirected. In fact, a great majority of existing models assumes that the DM particle has spin 0, 1/2, or 1, much as the known particles of the SM. There are no-go theorems that forbid *massless* elementary particles with spin higher than two [16,17] and strongly constrain massless spin-3/2 and spin-2 theories.[1] However, there is no obstruction for a consistent EFT with massive higher-spin particles [18].[2] In this vein, dark matter with spin $S > 1$ can be realized in the framework of an EFT valid only for energies below some cutoff scale $\Lambda$. Among this class of models, the most studied case by far is the spin-3/2 DM (see e.g. [23–29]), motivated largely by the gravitino in the supergravity extensions of the SM. Spin-2 DM can be realized in the context of bigravity [30–32], and spin-3 DM [33, 34] was also considered. DM with arbitrarily large spin can arise in the framework of large $N$ gauge theories. In particular, Ref. [35] studied a scenario where the DM particle is a baryon of a confining $SU(N_c)$ dark sector. In this case the lightest dark baryon corresponds to the totally symmetric spin configuration, therefore $S = N_c/2$. Very recently, DM of generic spin $S$ was explored for a Higgs-portal scenario [36] and for gravitational DM production during inflation [37].

In this paper, we solve a technical difficulty that impedes the construction of higher-spin

---

[1]In this paper we are concerned with Poincaré-invariant, unitary, local QFTs in four spacetime dimensions. If one of these assumptions is lifted, massless higher-spin particles may be allowed.

[2]Unitarity and causality impose severe restrictions on the spectrum and interactions of such theories [15,19,20], but these bounds can be avoided if the EFT cutoff is not much higher than the mass scale of the higher-spin particles. Positivity [21, 22] may also provide strong constraints, as shown for quartic interactions of a spin-3 particle in Ref. [18]. However, positivity bounds will not be relevant for the interactions considered in this paper, as it is unclear whether such bounds apply for gravitational couplings.

DM models and performing efficient calculations of observables. We completely bypass fields and Lagrangians (which are notoriously messy for higher spins and introduce complications due to unphysical degrees of freedom), and instead employ the on-shell amplitude techniques.[3] A DM model is defined via its on-shell 3-point amplitudes, which are written down using the massive spinor formalism of Ref. [15]. Starting from these, the relevant 2-to-2 scattering amplitudes with the DM particle(s) are constructed by gluing the 3-point ones according to the rules of unitarity and locality. We will show that this procedure greatly simplifies dealing with higher-spin DM theories. Not only the rates of relevant process can be efficiently calculated, but also the physics is transparent, in particular the overall energy/temperature dependence can be quickly obtained without any calculations using simple dimensional analysis. Finally, the validity regime of the EFT can be readily estimated by studying the energy dependence of several processes (annihilation, Compton scattering, self-scattering), to ensure that the calculations are consistent with perturbative unitarity.

In order to illustrate the utility of on-shell methods in this context, we study two simple scenarios with higher-spin DM. We first consider a scenario where the DM particle $X$ has only minimal coupling to gravity. DM stability is guaranteed by a $\mathbb{Z}_2$ symmetry, which also forbids direct 3-point couplings of $X$ to the SM. In this framework, we focus on the freeze-in DM production from a thermalized SM bath, assuming that at some temperature $T_{\max}$ all SM particles are thermalized and the initial DM abundance is negligible. The on-shell formalism allows us to easily compute the annihilation rate of the SM particles into a spin-$S$ DM mediated by the massless graviton. It also allows us to study the deformations of the gravity-mediated amplitude by 4-point contact interactions, which can modify the overall energy behaviour of the amplitude. We find that, even in this truly minimal scenario, the freeze-in DM abundance can match the experimentally observed one for a broad range of DM masses, from sub-TeV to GUT-scale masses. This is thanks to the fact that, for higher-spin particles, the gravitational interaction strength of longitudinal components grows with energy $E$ quicker than $E^2/M_{\text{Pl}}^2$. Next, we relax the $\mathbb{Z}_2$ symmetry assumption and allow the DM to couple to the SM. We restrict to the scenario where $X$ couples only to a pair of Higgs doublets, and does not have direct 3-point couplings with the remaining SM particles. In this scenario new production processes should be taken into account, in particular Higgs-mediated pair production, as well as single production in association with an electroweak boson or a top quark. At the same time, DM stability is no longer protected by $\mathbb{Z}_2$, and we have to deal with stringent constraints on decaying DM. We again show that one can obtain the correct DM abundance while avoiding the decay constraints. However, the parameter space where the freeze-in production qualitatively departs from the gravity-mediated scenario (and is instead dominated by single production) is limited to a narrow range of fairly light DM masses.

The paper is organized as follows. The freeze-in mechanism of DM production is briefly reviewed in Section 2. The gravity-mediated scenario is explored in Section 3, in which we present the annihilation amplitudes for a generic spin-$S$ DM. In Section 4, we allow the DM to have a direct coupling to the SM through a $\mathbb{Z}_2$-violating DM-Higgs-Higgs amplitude. We reserve Section 5 for conclusions and future directions. A short summary of the massless and massive spinor formalism can be found in Appendix A and the derivation of the scattering amplitudes used in the paper is presented in Appendix B.

---

[3]A different Lagrangian-less approach to higher-spin calculations is pursued in Ref. [36], based on the formalism developed in Ref. [38].

## 2 Freeze-in recap

In this article we focus on dark matter production via the freeze-in mechanism [39–41]. This occurs when particles in the thermal bath can annihilate into dark matter but the inverse process can be neglected due to feeble interactions and/or low number density of dark matter. Freeze-in is relevant in our scenario, as our higher-spin dark matter couples to the SM only through irrelevant operators suppressed by powers of a high scale.

We assume that the production occur in the radiation-dominated era and the thermal bath consists of only SM particles at a common temperature $T$. The goal is to calculate the yield $Y_X = n_X/s$, computed as a function of $T$, where $n_X$ is the number density of the dark matter particle $X$, and $s$ is the entropy density. The evolution equation reads

$$-THs\frac{dY_X}{dT} = R_X,\tag{1}$$

where $s = \frac{2\pi^2}{45}g_* T^3$, the Hubble rate $H = \sqrt{g_*}\pi T^2/\sqrt{90}\,M_{\mathrm{Pl}}$, $M_{\mathrm{Pl}} = 2.44 \times 10^{18}$ GeV, and $g_* = 106.75$ when all the SM degrees of freedom are in equilibrium with the thermal bath. On the right-hand side, $R_X$ encodes the production rate, which depends on the details of the interactions between dark matter and SM. We assume that $\psi\chi \to \phi X$ and $\psi\chi \to XX$ annihilation processes dominate the production, where $\psi$, $\chi$, $\phi$ stand for SM particles. The amplitude for this process is denoted $\mathcal{M}_{\mathrm{ann}}$. Ignoring the inverse annihilation and Bose enhancement or Pauli blocking, the production rate is given by

$$R_X \equiv \int d\Phi f_\psi f_\chi |\mathcal{M}_{\mathrm{ann}}|^2,\tag{2}$$

where $f_i$ is the distribution function for a given SM species, and $d\Phi$ is the phase space element, $d\Phi \equiv (2\pi)^4\delta^4(p_\psi + p_\chi - k_1 - k_2)d\Phi(p_\psi)d\Phi(p_\chi)d\Phi(k_1)d\Phi(k_2)$, $d\Phi(p_i) \equiv \frac{d^3 p_i}{2E_i(2\pi)^3}$, $E_i \equiv \sqrt{\mathbf{p_i}^2 + m_i^2}$. This expression is valid regardless whether the final state contains one or two $X$'s.[4] The amplitude square is implicitly summed/averaged over all internal degrees of freedom of $\psi$ and $\chi$, such as color or spin. If several distinct annihilation processes contribute significantly to the number density of $X$, then $R_X$ should be summed over these processes.

We assume the SM particles can be treated as massless (that is, $T \gtrsim 1$ TeV), and approximate $f_i$ by the equilibrium Maxwell distribution, $f_i = g_i e^{-E_i/T}$, where $g_i$ is the number of internal degrees of freedom. Then (2) can be simplified to [42]

$$R_X = \frac{T}{64\pi^4}\int_{s_0}^\infty ds\, \beta \sqrt{s}\, K_1\left(\frac{\sqrt{s}}{T}\right)I(s),\tag{3}$$

where $K_1$ is the modified Bessel function of the second kind, $s = (p_\psi + p_\chi)^2$, $s_0 = 4m^2$ and $\beta = \sqrt{1 - 4m^2/s}$ for annihilation into $XX$, while $s_0 = m^2$ and $\beta = 1 - m^2/s$ for annihilation into $\phi X$, where $m$ is the mass of the dark matter particle $X$. The function $I(s)$ describes the matrix element squared integrated over the 2-body phase space, and can be expressed as

$$I(s) = \frac{g_\psi g_\chi}{16\pi}\int_{-1}^1 d\cos\theta\, |\mathcal{M}_{\mathrm{ann}}|^2,\tag{4}$$

where $\theta$ is the angle between $\psi$ and $X$ in the center of mass frame.

---

[4]If two $X$'s are produced in the annihilation, then $R_X$ should contain a factor of two in front, which however cancels against $1/2!$ in the phase space, due to indistinguishable particles in the final state.

In the rest of this paper we will use Eq. (1) together with Eq. (3) to calculate the abundance of higher-spin dark matter in various scenarios. A priori we will be allowing a wide range for the dark matter mass:

$$5 \text{ keV} \lesssim m \lesssim M_{\text{Pl}}. \tag{5}$$

The lower end corresponds to the limits from small-scale structure [43], while the upper end is the general limit for elementary point-like particles. We will also require that the maximum temperature of the universe, denoted as $T_{\text{max}}$,[5] is within the validity regime of the EFT describing the DM+SM system. More precisely we impose the constraint

$$T_{\text{max}} < \alpha \Lambda, \tag{6}$$

where $\Lambda$ is the EFT cutoff, and $\alpha \ll 1$ in order to ensure perturbative control of the freeze-in calculation. We will demand that the integral in Eq. (3) is dominated by $\sqrt{s}$ below the cut-off, which typically requires $\alpha \sim \mathcal{O}(0.1)$ for the processes studied in this paper.

## 3 Gravity-mediated spin-S dark matter

In this section we discuss the most minimal scenario for higher-spin DM. We identify DM with a particle $X$ of mass $m$ and spin-$S > 1$, thus with $2S + 1$ degrees of freedom corresponding to different polarizations $P \in [-S, S]$. For the sake of this discussion, we assume $X$ is an isolated state with no other interactions than the minimal gravitational ones (which are required for all particles in nature). Direct 3-point coupling between DM and SM, as well as cubic DM self-interactions are forbidden by the $\mathbb{Z}_2$ symmetry acting as $X \to -X$. This is a particularly simple and economical model, with only a couple of adjustable parameters affecting the DM abundance in the universe.

We assume the DM particle is produced via the freeze-in process reviewed in Section 2 (we briefly comment on the freeze-out later in the text). In order to compute the relevant production amplitudes we will use the on-shell formalism. This leads to significant simplifications to the Lagrangian methods, which are notoriously messy for higher-spin particles. The basic objects from which we build our amplitudes are 2-component spinors denoted as $\chi_i^J$ and $\tilde{\chi}_i^J$, $(J = 1, 2)$ or $|\mathbf{i}\rangle$ and $|\mathbf{i}]$ in the short-hand notation. These are closely related to the more familiar spinor wave functions for spin-1/2 fermions. Given the momentum $p_i^\mu$ of the $i$-th particle, the corresponding spinors can be determined from the equation $p_i \sigma = \sum_{J=1}^2 \chi_i^J \tilde{\chi}_{iJ}$ up to little group transformations acting on the SU(2) indices $J$. Amplitudes are composed of Lorentz-invariant contractions $\chi_i \chi_j \equiv \langle \mathbf{ij} \rangle$ and $\tilde{\chi}_i \tilde{\chi}_j \equiv [\mathbf{ij}]$. More details about our conventions and a brief summary of the spinor-helicity formalism for massless and massive particles can be found in Appendix A. Given the 3-point amplitudes consistent with our assumptions, the production amplitudes are determined by demanding the correct factorization on the kinematics poles, as required by unitarity and locality.

### 3.1 3-point amplitudes

The minimal coupling between the graviton and the spin-$S$ DM is described by the following on-shell 3-point amplitudes [15]:

$$\mathcal{M}(\mathbf{1}_X \mathbf{2}_X 3_h^-) = -\frac{(A_{12}^-)^2}{M_{\text{Pl}}} \frac{[\mathbf{21}]^{2S}}{m^{2S}}, \qquad \mathcal{M}(\mathbf{1}_X \mathbf{2}_X 3_h^+) = -\frac{(A_{12}^+)^2}{M_{\text{Pl}}} \frac{\langle \mathbf{21} \rangle^{2S}}{m^{2S}}, \tag{1}$$

---

[5]In this paper we assume instantaneous reheating, identify $T_{\text{max}}$ with the reheating temperature, and avoid going into details of the reheating process. Going beyond the instantaneous reheating approximation, the relevant temperature for the DM production $T_{\text{max}}$ can be higher than the reheating temperature [44, 45], which may be numerically important when DM production rates, as in our scenario, rapidly increase with the temperature.

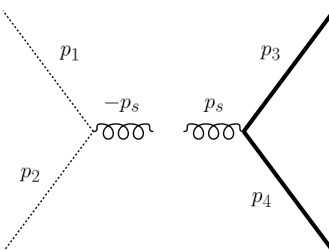

Figure 1: DM pair production in the gravity-mediated scenario. The thick line is the spin-$S$ DM, the dotted line is a SM particle and the curly line is the graviton. All momenta are incoming, and $p_s \equiv p_1 + p_2$.

where $A_{12}^{+} \equiv \langle \zeta p_1 3 ] / \langle 3 \zeta \rangle$ and $A_{12}^{-} \equiv [\zeta p_1 3 \rangle / [3 \zeta]$, with arbitrary reference spinors $|\zeta\rangle$ and $|\zeta]$. Due to the $\mathbb{Z}_2$ symmetry, couplings between one dark matter particle and two gravitons are forbidden, which also guarantees DM stability.

As for the SM matter, we approximate it as massless, that is we assume $T$ is much above the electroweak scale. Then its minimal interactions with the graviton are described by the following 3-point on-shell amplitudes:

$$\text{Spin } 0 : \qquad \mathcal{M}(1_\phi 2_\phi 3_h^\pm) = -\frac{(A_{12}^\pm)^2}{M_{\text{Pl}}},$$

$$\text{Spin } 1/2 : \qquad \mathcal{M}(1_\psi^- 2_{\bar\psi}^+ 3_h^\pm) = -\frac{A_{12}^\pm B_{12}^\pm}{M_{\text{Pl}}},$$

$$\text{Spin } 1 : \qquad \mathcal{M}(1_v^- 2_v^+ 3_h^\pm) = -\frac{(B_{12}^\pm)^2}{M_{\text{Pl}}}, \qquad (2)$$

where $B_{12}^{+} \equiv \langle 1 \zeta \rangle [23] / \langle 3 \zeta \rangle$ and $B_{12}^{-} \equiv \langle 13 \rangle [2\zeta] / [3\zeta]$, with arbitrary reference spinors $|\zeta\rangle$ and $|\zeta]$.

## 3.2 DM production amplitudes

In this scenario DM is produced via annihilation of two SM particles into two DM particles (schematically on Fig. 1). Independently of the SM and DM spins, the corresponding amplitudes have only $s$-channel poles corresponding to the massless graviton exchange. We find the amplitudes describing annihilation of SM matter into spin-$S$ dark matter have the form

$$\mathcal{M}(1_\phi 2_{\bar\phi} 3_X 4_X) = \frac{1}{s M_{\text{Pl}}^2 m^{2S-1}} \left\{ \frac{t-u}{2} \left( \langle 3p_1 4 ] + \langle 4p_1 3 ] \right) \sum_{k=1}^{2S-2} [43]^k \langle 43 \rangle^{2S-1-k} \right.$$
$$\left. - m \left( \langle 3p_1 4 ] + \langle 4p_1 3 ] \right)^2 \sum_{k=0}^{2S-2} [43]^k \langle 43 \rangle^{2S-2-k} \right\} + C_\phi, \qquad (3)$$

$$\mathcal{M}(1_\psi^- 2_{\bar\psi}^+ 3_X 4_X) = -\frac{1}{s M_{\text{Pl}}^2 m^{2S-1}} \left( \langle 31 \rangle [42] + \langle 41 \rangle [32] \right) \left\{ \frac{t-u}{2} \sum_{k=1}^{2S-2} [43]^k \langle 43 \rangle^{2S-1-k} \right.$$
$$\left. - m \left( \langle 3p_1 4 ] + \langle 4p_1 3 ] \right) \sum_{k=0}^{2S-2} [43]^k \langle 43 \rangle^{2S-2-k} \right\} + C_\psi, \qquad (4)$$

$$\mathcal{M}(1_\nu^- 2_{\bar{\nu}}^+ 3_X 4_X) = -\frac{1}{s\,M_{\text{Pl}}^2 m^{2S-1}} \Bigg\{ \langle 1 p_3 2](\langle 31\rangle[42] + \langle 41\rangle[32]) \sum_{k=1}^{2S-2} [43]^k \langle 43\rangle^{2S-1-k}$$

$$+ \quad m(\langle 31\rangle[42] + \langle 41\rangle[32])^2 \sum_{k=0}^{2S-2} [43]^k \langle 43\rangle^{2S-2-k} \Bigg\} + C_\nu, \tag{5}$$

where $\phi$, $\psi$, $\nu$ denote SM particles of spin 0, 1/2, 1, respectively. For the sake of the discussion in this section we assume $S \geq 2$, but notice that these amplitudes are also valid as they stand for $S = 3/2$.[6] The details of the calculation are shown in Appendix B. In the above, $C_\phi$, $C_\psi$, and $C_\nu$ are contact terms without poles in any kinematic channel. The leading contact terms in the EFT expansion are

$$C_\phi = \frac{1}{M_{\text{Pl}}^2 m^{2S-2}} \sum_{k=0}^{2S} \mathcal{C}_\phi^{(k+1)} [43]^k \langle 43\rangle^{2S-k} + \cdots,$$

$$C_\psi = \frac{1}{M_{\text{Pl}}^2 m^{2S-1}} (\langle 31\rangle[42] - \langle 41\rangle[32]) \sum_{k=0}^{2S-1} \mathcal{C}_\psi^{(k+1)} [43]^k \langle 43\rangle^{2S-1-k} + \cdots,$$

$$C_\nu = \frac{1}{M_{\text{Pl}}^2 m^{2S}} \langle 31\rangle\langle 41\rangle[32][42] \sum_{k=0}^{2S-2} \mathcal{C}_\nu^{(k+1)} [43]^k \langle 43\rangle^{2S-2-k} + \cdots. \tag{6}$$

These contact terms can be classified in two different kinds. On one hand, the contact terms that do not change the overall energy behavior, leading to a scenario completely equivalent to the one without contact terms. On the other hand, contact terms that change the high energy dependence resulting in a scenario dominated by contact interaction.[7] For spin-1/2 and spin-1 matter one could also consider the amplitudes $\mathcal{M}(1_f^\pm 2_f^\pm 3_X 4_X)$, $f = \psi, \nu$, which are pure contact terms. Notice that in our normalization the Wilson coefficients scale as $\mathcal{C}_{\phi,\psi,\nu}^{(i)} \sim g_*^2 M_{\text{Pl}}^2/\Lambda^2$, where $\Lambda$ is the mass scale of the UV completion producing these contact terms, and $g_*$ is the coupling strength of the new degrees of freedom.[8] For this reason, large values of $\mathcal{C}_{\phi,\psi,\nu}^{(i)}$ are consistent with the EFT expansion if new physics enters below the Planck scale. In particular, $\Lambda \sim \mathcal{O}(\text{TeV})$ and $g_* \sim 1$ correspond to $\mathcal{C}_{\phi,\psi,\nu}^{(i)} \sim 10^{30}$.

For most of the following discussion we will ignore the contact terms, and focus on the s-channel graviton pole contributions in Eqs. (3)-(5). We will comment on the effects of the contact terms later in Section 3.5.

## 3.3 Dimensional analysis

Before we plunge into numerical analysis of DM production in our scenario, let us first establish some basic intuitions using simple dimensional analysis. For $s \gg 4m^2$, the annihilation amplitudes in Eqs. (3)-(5) behave as

$$\mathcal{M}_{\text{ann}} \sim \frac{E^{2S+1}}{M_{\text{Pl}}^2 m^{2S-1}}. \tag{7}$$

---

[6]Eqs. (3)-(5) are also valid for $S = 1$ with the convention that the first summation in each amplitude should be replaced by zero. They are not valid for $S = 0$ and $S = 1/2$.

[7]Operators of higher dimension cannot be obtained simply attaching powers of Mandelstam variables in the minimum spinor structures. Mass induced identities can relate apparently independent structures and a more careful analysis is required to determine the basis of independent contact terms [10].

[8]The $g_*$ scaling follows from the fact that $\mathcal{C}_{\phi,\psi,\nu}^{(i)}$ is order $\hbar^{-1}$ in the $\hbar$ counting (see e.g. [46]) and the assumption that the contact term is generated at tree level. We work in the normalization where the gauge and Yukawa couplings are $g_* \sim \hbar^{-1/2}$.

This scaling is valid for $S \geq 3/2$ as long as we ignore the contact terms $C_i$ (for $S = 1$ one instead has $\mathcal{M}_{\text{ann}} \sim E^2/M_{\text{Pl}}^2$). Eq. (7) results in the annihilation rate and DM yield

$$T_{\text{max}} \gg m \quad \to \quad R_X \sim \frac{T^{4S+6}}{M_{\text{Pl}}^4 m^{4S-2}} \quad \to \quad Y_X^0 \sim \frac{T_{\text{max}}^{4S+1}}{M_{\text{Pl}}^3 m^{4S-2}}. \tag{8}$$

Matching that to the measured dark matter abundance, the required maximal temperature of the universe to match the observations is given by

$$T_{\text{max}} \gg m \quad \to \quad T_{\text{max}} \sim (y_{\text{ref}} M_{\text{Pl}}^3 m^{4S-3})^{\frac{1}{4S+1}}, \tag{9}$$

where $y_{\text{ref}} = \rho_c \Omega_X/s_0 \approx 4.1 \cdot 10^{-10}$ GeV. For very large DM masses, $m \gtrsim (y_{\text{ref}} M_{\text{Pl}}^3)^{1/4} \sim 10^{11}$ GeV, the estimate in Eq. (9) returns $T_{\text{max}} \lesssim m$, in contradiction to the assumption $T_{\text{max}} \gg m$ used in the derivation. In this regime the estimate of the required maximal temperature has to be modified. For $T_{\text{max}} \ll m$, DM is produced with small non-relativistic velocities from the SM particle at the tail of the thermal distribution. Due to the Boltzmann suppression we have

$$T_{\text{max}} \ll m \quad \to \quad R_X \sim \frac{m^8}{M_{\text{Pl}}^4} e^{-2m/T} \quad \to \quad Y_X^0 \sim \frac{m^3}{M_{\text{Pl}}^3} e^{-2m/T_{\text{max}}}, \tag{10}$$

independently of the DM spin[9]. Therefore the required maximal temperature is estimated as

$$T_{\text{max}} \ll m \quad \to \quad T_{\text{max}} \sim \frac{2m}{\log\left(\frac{m^4}{y_{\text{ref}} M_{\text{Pl}}^3}\right)}. \tag{11}$$

The annihilation amplitude grows with energy and hits the strong coupling at $E \simeq \Lambda_a$, where $\Lambda_a$ can be estimated as

$$\Lambda_a \sim (\sqrt{4\pi} M_{\text{Pl}} m^{S-1/2})^{\frac{1}{S+1/2}}. \tag{12}$$

Furthermore, amplitudes for self- and gravitational Compton scattering processes grow even faster for $s \gg m^2$. The associated strong coupling scale can be estimated as [48]

$$\mathcal{M}_{\text{self}} \sim \mathcal{M}_{\text{Compton}} \sim \frac{E^{4S-2}}{M_{\text{Pl}}^2 m^{4S-4}} \quad \longrightarrow \quad \Lambda_s \sim (\sqrt{4\pi} M_{\text{Pl}} m^{2S-2})^{\frac{1}{2S-1}}. \tag{13}$$

We have $\Lambda_s \leq \Lambda_a$ for any $S \geq 3/2$, thus $\Lambda_s$ sets the maximum possible value of the EFT cut-off scale $\Lambda$ in our scenario. We can see that, as $S$ is increased, $\Lambda_s$ approaches $m$ due to the amplitudes' steeper growth with energy. This indicates that, in the regime $T_{\text{max}} \gg m$ where freeze-in production is dominated at $\sqrt{s} \sim 20m$, perturbative control of the calculation will be lost for large enough $S$. Indeed, the condition in Eq. (6) together with Eq. (9) can be used to derive the lower limit on the DM mass:

$$m > \left[ \frac{M_{\text{pl}}^{2S-4} y_{\text{ref}}^{2S-1}}{\alpha^{(4S+1)(2S-1)} (4\pi)^{2S+1/2}} \right]^{\frac{1}{4S-5}} \overset{S \gg 2}{\to} \alpha^{-2S} \left[ \frac{M_{\text{pl}} y_{\text{ref}}}{4\pi} \right]^{1/2} \approx \alpha^{-2S} \times 30 \text{ TeV}. \tag{14}$$

Using $\alpha \sim 0.05$, that is requiring that $\Lambda_s$ is a factor of 20 above $T_{\text{max}}$ to ensure perturbative control over the freeze-in calculation, we find that already for $S > 3$ the lower limit on DM exceeds $(y_{\text{ref}} M_{\text{Pl}})^{1/4} \sim 10^{11}$ GeV, in which case the allowed region is fully contained in the $T_{\text{max}} \ll m$ regime. Thus, DM with $S > 3$ is phenomenologically viable only in the large $m$ region where DM is produced with non-relativistic velocity and spin plays a minor role in the freeze-in dynamics. In the $T_{\text{max}} \ll m$ regime DM is produced right at the threshold where physics is always perturbative for $m \lesssim M_{\text{Pl}}$.

---

[9]More precisely, one finds $R_X \sim \frac{m^5 T^3}{M_{\text{Pl}}^4} e^{-2m/T}$ for bosonic DM, and $R_X \sim \frac{m^4 T^4}{M_{\text{Pl}}^4} e^{-2m/T}$ for fermionic DM of any spin, the difference being due to $\mathcal{M}_{\text{ann}}$ vanishing on the DM production threshold for fermions. In this regime power corrections in $T$ affect only logarithmically the estimate of $T_{\text{max}}$ in Eq. (11) and are ignored here. Moreover, we ignore here possible effects due to bound state formation [47].

### 3.4 Quantitative results

For a concrete value of the DM spin-$S$ it is straightforward to calculate the freeze-in rate $R_X$ in Eq. (2) starting from the annihilation amplitudes in Eqs. (3)-(5), which should be substituted into Eq. (3) to obtain the production rate. Summing over the SM states in the thermal bath, and numerically calculating the evolution of the DM yield $Y_X$, one obtains the DM relic abundance as a function of the DM mass $m$ and the maximum temperature of the universe $T_{\max}$. Given this expression, we adjust $T_{\max}$ so as to reproduce the experimentally observed DM yield, $Y_X \approx 4.1 \times 10^{-10} \text{GeV}/m$. Formally, that adjustment is always possible but for some regions of the $m$-$T_{\max}$ parameter space the perturbative control over the freeze-in calculation is lost, and we treat those regions as excluded.

We start this discussion with spin $S = 2$ DM, assuming the absence of contact terms $C_i$ in Eqs. (3)-(5). Following the algorithm described above, we find that the production rate of DM produced from annihilation of (complex) SM scalars, spin-1/2 fermion, and spin-1 vectors in the limit $T \gg m$ is given by

$$R_\phi \approx \frac{960}{\pi^5 m^6 M_{\text{Pl}}^4} T^{14}, \qquad R_\psi \approx \frac{3840}{\pi^5 m^6 M_{\text{Pl}}^4} T^{14}, \qquad R_\nu \approx \frac{11520}{\pi^5 m^6 M_{\text{Pl}}^4} T^{14}. \qquad (15)$$

Summing that over all degrees of freedom of the SM, the total rate is

$$R_X = 2 \times R_\phi + 45 \times R_\psi + 6 \times R_\nu \approx \frac{243840}{\pi^5 m^6 M_{\text{Pl}}^4} T^{14}, \qquad (16)$$

which leads to the DM yield for $T_{\max} \gg m$:

$$Y_X^0 \approx \frac{0.55}{m^6 M_{\text{Pl}}^3} T_{\max}^9. \qquad (17)$$

This agrees well with the dimensional estimate in Eq. (9), up to a numerical factor that has little practical relevance, especially in comparison with the steep dependence on $T_{\max}$. On the other hand, for $T_{\max} \lesssim m$ the result in Eq. (17) is not valid. In this regime, the DM yield can be approximated by

$$Y_X^0 \approx \frac{2.7 \times 10^{-4} \, m^4 + 1.2 \times 10^{-3} \, m^3 T_{\max} + 2.0 \times 10^{-2} \, m^2 T_{\max}^2}{M_{\text{Pl}}^3 T_{\max}} e^{-\frac{2m}{T_{\max}}}, \qquad (18)$$

which agrees with the dimensional estimate in Eq. (11), up to numerical factors and power corrections in $T_{\max}/m$. Again, these power corrections have little practical relevance, as $T_{\max}/m$ is $\mathcal{O}(0.1-1)$ in the phenomenologically viable parameter space.

The parameter space in the $S = 2$ scenario is displayed in Fig. 2. The solid black line marks the region where the relic abundance matches the observed one. For $m \lesssim 10^{13}$ GeV it is very well approximated by the simple estimate in Eq. (9), which is indicated by the dotted line. For larger $m$ the black line follows instead the estimate in Eq. (11). An important constraint on this scenario comes from requiring the DM production to be dominated by the energy range where the EFT is valid. For $S = 2$, the maximum possible cutoff of the EFT is set by the strong coupling scale of self-scattering and gravitational Compton scattering: $\Lambda_s = (\sqrt{4\pi} m^2 M_{\text{Pl}})^{1/3}$. As discussed earlier, for $T_{\max} \gg m$ we demand $T_{\max} < \alpha \Lambda_s$ with some $\alpha < 1$. Numerically we find that for $S = 2$ the production rate is dominated by $\sqrt{s} \sim 12m$, and becomes negligible for $\sqrt{s} \gtrsim 20m$, therefore we pick $\alpha = 1/20$ in this case. The parameter space excluded by this condition is marked orange in Fig. 2. We can see that the EFT validity bound is not overly restrictive in the $S = 2$ case, only excluding DM mass below $\sim 100$ MeV. All in all, for $S = 2$ the allowed range of DM masses is very generous:

$$S = 2: \qquad 100 \text{ MeV} \lesssim m \lesssim M_{\text{Pl}}, \qquad (19)$$



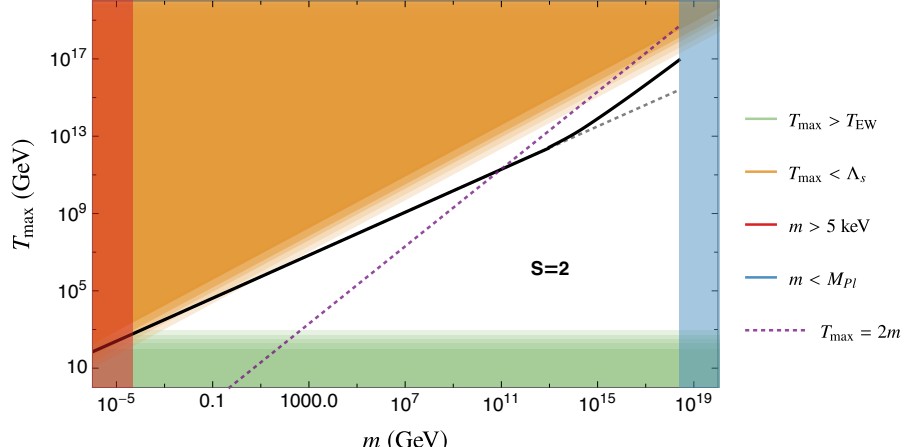

Figure 2: Spin-2 DM gravity-mediated scenario *without* contact terms. For a given DM mass $m$, the black line shows the maximum temperature of the universe needed to obtain correct relic abundance from freeze-in. The dashed-gray line indicates the simple estimate for $T_{\max}$ based on dimensional analysis, cf. Eq. (9). The lower bound on the DM mass, $m \gtrsim 5$ keV, is due to small-structure formation constraints, and the upper bound is the Planck scale. The orange region is excluded by perturbativity constraints on the EFT, requiring the separation between the strong coupling scale and $T_{\max}$ to be $\alpha = 1/20$. The green region corresponds to $T_{\max}$ below the electroweak scale.

corresponding to the range of $T_{\max}$:

$$\mathbf{S = 2}: \qquad 1 \text{ TeV} \lesssim T_{\max} \lesssim 10^{17} \text{ GeV}. \qquad (20)$$

Notice that the maximal temperature is always above the electroweak scale, justifying a posteriori approximating the SM degrees of freedom as massless.

Given the calculated production rate, we can also estimate the DM abundance assuming the standard freeze-out scenario, with DM in equilibrium with the SM. It turns out that then the correct abundance cannot be obtain in the region of parameters allowed by the perturbativity condition. This justifies the choice of an out-of-equilibrium DM production.

For higher spins the calculation follows analogous steps. The parameter space for $S = 5/2$ and $S = 3$ is shown in Fig. 3. Once again the region where the DM abundance matches the

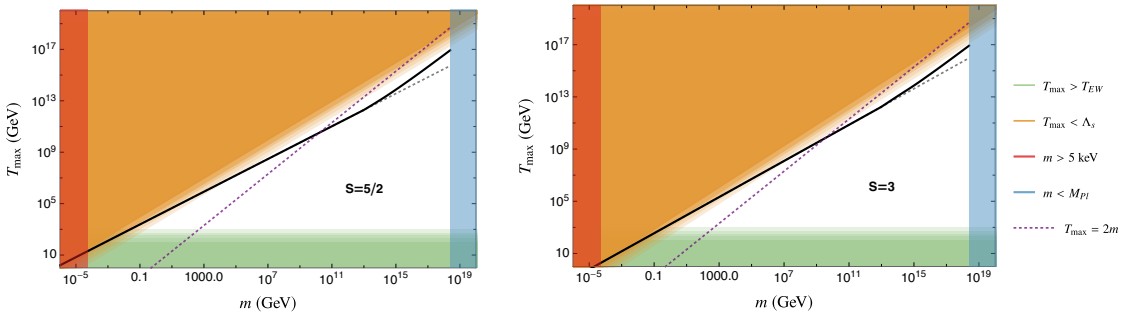

Figure 3: Spin-5/2 (left) and spin-3 (right) DM gravity-mediated scenario without contact terms. See the caption of Fig. 2 for explanations.

observed one follows the simple estimate in Eq. (9), deviating from it substantially only for very large DM masses. As anticipated in the previous subsection, the EFT validity bounds become more and more restrictive as we increase the DM spin. We find

$$
\begin{aligned}
\mathbf{S = 5/2}: &\qquad 10^2 \text{ TeV} \lesssim m \lesssim M_{\text{Pl}}, \\
\mathbf{S = 3}: &\qquad 10^4 \text{ TeV} \lesssim m \lesssim M_{\text{Pl}}.
\end{aligned}
\tag{21}
$$

For larger spins the phenomenologically viable DM production occurs only in the region where $T_{\text{max}} < 2m$, in which case it is produced with non-relativistic velocities and the dependence on DM spin is not substantial.

### 3.5 Effect of contact terms

The leading contact terms shown in Eq. (6) scale as

$$
C_\phi \sim \frac{E^{2S}}{M_{\text{Pl}}^2 m^{2S-2}}, \quad C_\psi \sim \frac{E^{2S+1}}{M_{\text{Pl}}^2 m^{2S-1}}, \quad C_v \sim \frac{E^{2S+2}}{M_{\text{Pl}}^2 m^{2S}}.
\tag{22}
$$

Given that the annihilation amplitudes in the absence of contact terms behave as $\mathcal{O}(E^{2S+1})$, we can see that for the case of SM vectors annihilating into DM the leading contact term may dominate the production if the Wilson coefficients $C_v$ are large enough. This modifies the DM yield to

$$
T_{\text{max}} \gg m \quad \rightarrow \quad R_X \sim \frac{T^{4S+8}}{M_{\text{Pl}}^4 m^{4S}} \quad \rightarrow \quad Y_X^0 \sim \frac{T_{\text{max}}^{4S+3}}{M_{\text{Pl}}^3 m^{4S}},
\tag{23}
$$

which modifies the required maximal temperature to

$$
T_{\text{max}} \gg m \quad \rightarrow \quad T_{\text{max}} \sim (y_{\text{ref}} M_{\text{Pl}}^3 m^{4S-1})^{\frac{1}{4S+3}}.
\tag{24}
$$

Adding contact terms that dominate the production has the effect of lowering the maximal temperature for a given mass in comparison with the case without the contact terms.

Let us study more carefully the concrete case of $S = 2$, which gives $\mathcal{M}_{\text{ann}} \sim C_v \sim E^6/m^4 M_{\text{Pl}}^2$. We include for simplicity a single contact term corresponding to $\mathcal{C}_v^{(2)}$ in Eq. (6):

$$
\frac{\mathcal{C}_v^{(2)}}{m^4 M_{\text{Pl}}^2} \langle 31 \rangle \langle 41 \rangle [32][42][43] \langle 43 \rangle.
\tag{25}
$$

The production rate is dominated by the contact term and the production rate and in the regime $T_{\text{max}} \gg m$ is given by

$$
R_X \approx 1290240 \, \frac{(\mathcal{C}_v^{(2)})^2}{\pi^5 m^8 M_{\text{Pl}}^4} T^{16} + 23040 \, \frac{3 + 10\,\mathcal{C}_v^{(2)} + (\mathcal{C}_v^{(2)})^2}{\pi^5 m^6 M_{\text{Pl}}^4} T^{14} + \frac{174720}{\pi^5 m^6 M_{\text{Pl}}^4} T^{14},
\tag{26}
$$

where the result of Eq. (16) is recovered by setting $\mathcal{C}_v^{(2)} = 0$. The resulting yield is

$$
Y_X^0 \approx 2.4 \, \frac{(\mathcal{C}_v^{(2)})^2}{m^8 M_{\text{Pl}}^3} T_{\text{max}}^{11} + \frac{0.5 + 0.5\,\mathcal{C}_v^{(2)} + 0.05\,(\mathcal{C}_v^{(2)})^2}{m^6 M_{\text{Pl}}^3} T_{\text{max}}^9.
\tag{27}
$$

The solutions of this equation evaluated for different maximal temperatures are shown on Fig. 4. We exclude the regions of the parameter space where $T_{\text{max}}$ falls below the electroweak symmetry breaking scale, to be consistent with our approximation of massless SM particles.

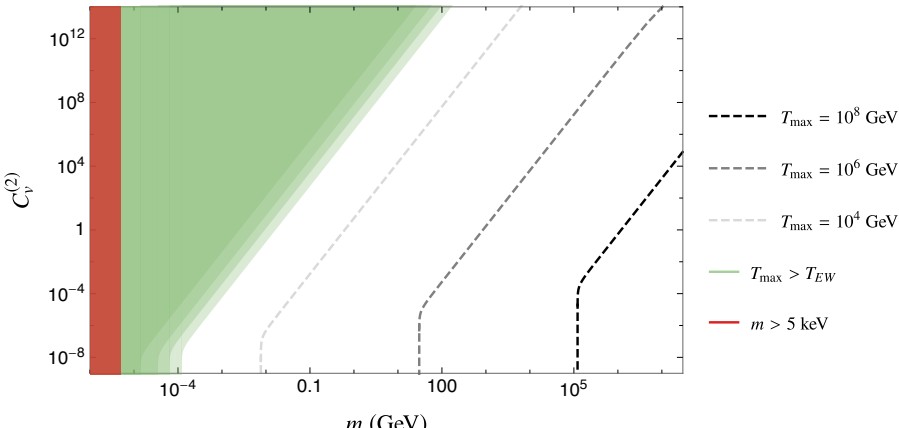

Figure 4: Contours of constant $T_{\max}$ leading to correct DM relic abundance as function of the DM mass $m$ and the magnitude $\mathcal{C}_v^{(2)}$ of the contact term in Eq. (25) for a spin-2 DM particle in the gravity-mediated scenario. The red region is excluded by the small-structure bound $m \gtrsim 5$ keV. The green region corresponds to $T_{\max}$ below the electroweak scale, contrary to the assumption used in our calculation. The right-bend of $T_{\max}$ contours marks the cross-over to the region where the contact term contribution dominates over the purely gravitational one.

The EFT perturbativity constraints on the maximum temperature of the universe have to be modified in this scenario to

$$T_{\max} \lesssim \alpha \min\left(\Lambda_s, \Lambda_C\right), \tag{28}$$

where $\Lambda_C \sim (4\pi m^4 M_{\rm Pl}^2/\mathcal{C}_v^{(2)})^{1/6}$. Notice that with the steeper energy growth of the production amplitude, $T_{\max}$ is lower than in the scenario without the contact term. This means that, in practice, the bound in Eq. (28) is always satisfied. As already anticipated, the addition of the contact term allows for a lower maximum temperature.

## 4 Spin-2 dark matter coupled to matter

In this section, we relax the assumption of a $\mathbb{Z}_2$ symmetry protecting the DM couplings. This allows the dark matter particle to have direct 3-point couplings to the SM sector. In the following we focus on the scenario where higher-spin DM interacts only with the SM Higgs doublet (in addition to gravitational interactions, of course). We also restrict the bulk of our discussion to spin $S = 2$ DM, and we will briefly comment on possible generalizations to $S > 2$. As in the previous section, we assume that freeze-in is the dominant process responsible for the relic abundance of DM. We continue approximating the SM particles as massless and working in the unbroken phase of the SM.

### 4.1 3-point amplitudes

We start with defining the 3-point amplitudes describing interactions of a spin-2 DM particle $X$. Regarding gravitational interactions, we continue assuming the minimal coupling, with the 3-point amplitudes given in Eq. (1) with $S = 2$. In addition, we allow for DM interacting with

the SM via the Higgs portal.[10] In our scenario, the interaction between DM and Higgs doublets is described by the on-shell 3-point amplitude:

$$\mathcal{M}(1_{H_a} 2_{\bar{H}_b} 3_X) = -\frac{c_H}{M_{\text{Pl}} m^2} \delta_{ab} \langle 3 p_1 3]^2. \tag{1}$$

In fact, the above describes a coupling of a massive spin-2 particle to the energy-momentum tensor of the Higgs doublet.[11] Here, $c_H$ is the order parameter for $\mathbb{Z}_2$-breaking which measures the DM-Higgs interaction strength. It scales as $c_H \sim \mathcal{O}(g_* M_{\text{Pl}}/\Lambda)$ (if the interaction is generated at tree level) or as $c_H \sim \mathcal{O}(g_*^3 M_{\text{Pl}}/16\pi^2\Lambda)$ (if the interaction is generated at one loop), where $\Lambda$ and $g_*$ are the mass scale and the coupling strength of the UV completion of our EFT. Assuming $g_* \sim 1$, $|c_H| \sim 1$ corresponds to a Planck-suppressed coupling, while $|c_H| \sim 10^{15}$ can be achieved if the UV completion enters at a TeV.

Given $\mathbb{Z}_2$ is broken by Eq. (1), nothing forbids a cubic self-interaction of DM. In our calculation we use the same 3-point amplitude as for the spin-2 massive graviton self-interaction in the DRGT gravity [50], which leads to scattering amplitudes that are maximally well-behaved in the high-energy limit [51]. Its spinor form is shown in Eq. (C.1) in Appendix C. For the sake of computing the DM annihilation amplitudes we also need the Higgs gauge and Yukawa couplings written in the spinor form. The former are

$$\mathcal{M}(1_{H_a} 2_{\bar{H}_b} 3_{v_c}^-) = -\sqrt{2} g_v T_{ab}^c \frac{\langle 13 \rangle \langle 23 \rangle}{\langle 12 \rangle}, \qquad \mathcal{M}(1_{H_a} 2_{\bar{H}_b} 3_{v_c}^+) = -\sqrt{2} g_v T_{ab}^c \frac{[13][23]}{[12]}. \tag{2}$$

For the SM $SU(2)$ gauge bosons $g_v = g$ and $T_{ab}^c = \sigma^c/2$ are the Pauli matrices, while for hypercharge gauge bosons $g_v = g'$ and $T_{ab}^Y = \frac{1}{2}\delta_{ab}$. For the Yukawa interactions we focus on the ones with the top quark.

$$\mathcal{M}(1_{H_a} 2_{Q_b}^- 3_{\bar{t}_R}^-) = -y_t \epsilon_{ab} \langle 23 \rangle, \qquad \mathcal{M}(1_{H_a} 2_{Q_b}^+ 3_{\bar{t}_R}^+) = -y_t \epsilon_{ab} [23], \tag{3}$$

where $Q$ denotes the 3rd generation left-handed quark doublet.

## 4.2 DM production amplitudes

The relevant processes for freeze-in DM production can be divided into *pair* and *single* production. The former includes a contribution with the massless graviton in the s-channel, cf. Fig. 1, which featured also in the gravity-mediated scenario discussed in the previous section and was given in Eqs. (3)-(5). A new element here is $H\bar{H} \to XX$ mediated either by a $t/u$-channel Higgs or by an s-channel DM, see Fig. 5 left. The $t/u$-channel piece can be approximated as

$$\mathcal{M}(1_{H_a} 2_{\bar{H}_b} 3_X 4_X) \quad = -\delta_{ab} \frac{c_H^2}{M_{\text{Pl}}^2 m^3} \left\{ \langle 3 p_1 3] \langle 4 p_2 4] \frac{\langle 34 \rangle \langle 3 p_1 4] + [34] \langle 4 p_1 3]}{t} \right.$$
$$\left. - \langle 3 p_2 3] \langle 4 p_1 4] \frac{\langle 34 \rangle \langle 4 p_1 3] + [34] \langle 3 p_1 4]}{u} + \mathcal{O}(m) \right\}. \tag{4}$$

The derivation is given in Appendix B.2. The s-channel piece is possible in the presence of cubic DM self-interactions. In Ref. [48] it was shown that it leads to the high-energy behavior as the $t/u$-channel piece when the self-interaction has the DRGT form in Eq. (C.1).[12] We do not show it explicitly here as it will play a minor role in the following discussion.

---

[10]A different Higgs-portal scenario with spin-$S$ DM was recently studied in Ref. [36], where the interaction is mediated by a *quartic* coupling between two Higgs doublets and two DM particles.

[11]The same coupling is present in the bi-gravity scenario of Ref. [49], however in this case DM couples universally to all SM particles, leading to a different phenomenology.

[12]We note also that for suitably chosen parameters $a_0$ and $a_2$ in Eq. (C.1) there can be cancellations between the $s$- and $t/u$-channel pieces, which could soften the high-energy behavior of $\mathcal{M}(1_{H_a} 2_{\bar{H}_b} 3_X 4_X)$ [48] and make this process (even) less relevant for freeze-in.

Pair production

Single production

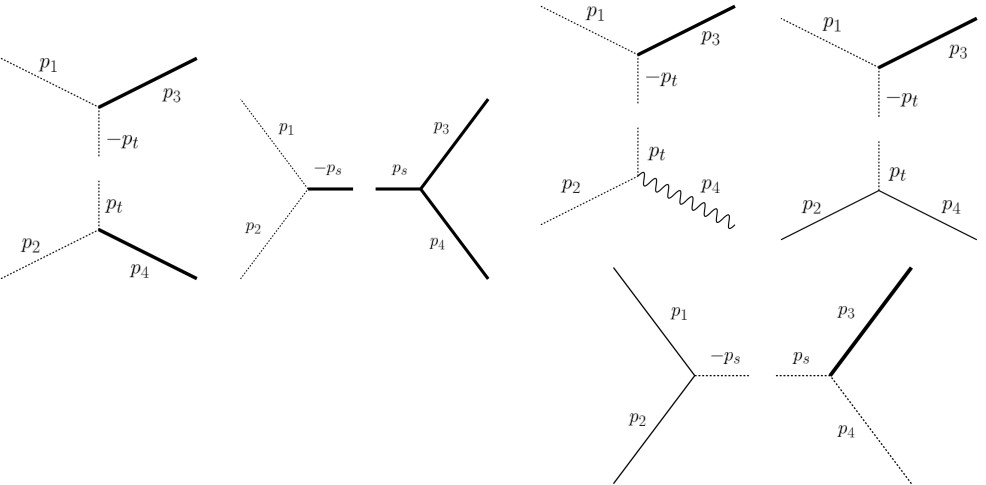

Figure 5: Schematic representation of Higgs-mediated contributions to DM production. The DM particle is depicted by a thick line, the Higgs doublet by a thin dotted line, and fermions by a thin solid line. All momenta are incoming, and we define $p_s = p_1 + p_2$, $p_t = p_1 + p_3$. For pair production, a u-channel contribution related by crossing to the t-channel one should be included as well.

Furthermore, the $\mathbb{Z}_2$-violating interaction in Eq. (1) opens the possibility of single DM production. There are several processes in this category. One is $H\bar{H} \to Xv$ with the Higgs in the $t/u$ channel, where $v$ stands for an electroweak gauge boson. The amplitude is given by

$$\mathcal{M}(1_{H_a} 2_{\bar{H}_b} 3_X 4_v^-) = \frac{c_H g_v T_{ab}^c}{\sqrt{2}\, M_{\text{Pl}} m^2}\left[ \frac{\langle 4p_1 p_2 4\rangle}{tu}([\mathbf{3}p_1\mathbf{3}]^2 + [\mathbf{3}p_2\mathbf{3}]^2 + [\mathbf{3}p_4\mathbf{3}]^2) + \right.$$
$$\left. + 2\langle 43\rangle\left(\frac{\langle 4p_2\mathbf{3}][\mathbf{3}p_2\mathbf{3}]}{t} - \frac{\langle 4p_1\mathbf{3}][\mathbf{3}p_1\mathbf{3}]}{u}\right)\right], \tag{5}$$

$$\mathcal{M}(1_{H_a} 2_{\bar{H}_b} 3_X 4_v^+) = \frac{c_H g_v T_{ab}^c}{\sqrt{2}\, M_{\text{Pl}} m^2}\left[ \frac{[4p_1 p_2 4]}{tu}([\mathbf{3}p_1\mathbf{3}]^2 + [\mathbf{3}p_2\mathbf{3}]^2 + [\mathbf{3}p_4\mathbf{3}]^2) + \right.$$
$$\left. + 2[43]\left(\frac{\langle \mathbf{3}p_2 4][\mathbf{3}p_2\mathbf{3}\rangle}{t} - \frac{\langle \mathbf{3}p_1 4][\mathbf{3}p_1\mathbf{3}\rangle}{u}\right)\right]. \tag{6}$$

The details of the derivation are given in Appendix B.3. Another relevant process is $Hv \to XH$, and its amplitude can be obtained from Eq. (5) by crossing. The other processes are $\psi\psi' \to XH$ with the Higgs in the s-channel, and $H\psi \to X\psi'$ with the Higgs in the $t/u$ channels, where $\psi$ and $\psi'$ stand for a 3rd generation quark doublet or singlet. For the former, the amplitudes up to contact terms take the form

$$\mathcal{M}(1_{Q_a}^- 2_{\bar{t}_R}^- 3_X 4_{H_b}) = \epsilon_{ab}\frac{y_t\, c_H}{M_{\text{Pl}} m^2 s}\langle 12\rangle\langle \mathbf{3}p_4\mathbf{3}]^2, \quad \mathcal{M}(1_{\bar{Q}_a}^+ 2_{t_R}^+ 3_X 4_{H_b}) = \epsilon_{ab}\frac{y_t\, c_H}{M_{\text{Pl}} m^2 s}[12]\langle \mathbf{3}p_4\mathbf{3}]^2. \tag{7}$$

In this case the derivation is trivial: the residue of the s-channel is obtained by simply gluing the corresponding 3-point amplitudes. The amplitudes for $Ht_R \to XQ$ and $HQ \to Xt_R$ can be obtained from Eq. (7) by crossing.

### 4.3 Dimensional analysis

Before we give the quantitative results for this model, we first analyze using dimensional analysis which of the processes discussed in the previous section dominates the DM production. The single and pair production amplitudes scale at high energies as

$$\mathcal{M}_{1\text{DM}} \sim \frac{c_H E^3}{M_{\text{Pl}} m^2}, \qquad \mathcal{M}_{2\text{DM}} \sim \left(1 + c_H^2\right)\frac{E^5}{M_{\text{Pl}}^2 m^3}, \tag{8}$$

where we omit the dependence on the SM gauge and Yukawa couplings and other $\mathcal{O}(1)$ coefficients. The pair production is suppressed by more powers of $M_{\text{Pl}}$, therefore single production will dominate, unless $c_H$ is very large or very small. In the limit $T_{\text{max}} \gg m$, these two contributions result in the DM yield

$$T_{\text{max}} \gg m \quad \rightarrow \quad Y_X^0 \sim \frac{T_{\text{max}}^5}{M_{\text{Pl}} m^4}\left[c_H^2 + \frac{T_{\text{max}}^4}{M_{\text{Pl}}^2 m^2}(1 + c_H^2)^2\right]. \tag{9}$$

Single production dominates when $T_{\text{max}} \lesssim \frac{|c_H|}{1+c_H^2}\sqrt{m\,M_{\text{Pl}}}$, while for larger temperatures the pair production prevails. Matching $Y_X^0$ in Eq. (9) to the observed DM abundance, $Y_X^0 = y_{\text{ref}}/m$, we can estimate $T_{\text{max}}$ for a given $m$ and $c_H$ and verify whether it falls into single or pair production domination region. The results of this investigation are shown in Fig. 6, where in the yellow region the gravity-mediated pair production dominates, and the scenario reduces to the one discussed in Section 3. The coefficient multiplying the DM-Higgs interaction in Eq. (1) is of order $|c_H| \sim g_* M_{\text{Pl}}/\Lambda$ so, barring transplanckian UV completions, the requirement $|c_H| \ll 1$ translates to a very weakly interacting UV completion. On the other hand, the Higgs-mediated pair production becomes relevant only for values of $|c_H| \gtrsim 10^{70}$, which cannot be achieved within the validity of our EFT assuming $\Lambda \gtrsim 1$ TeV. Therefore, we do not show this region in Fig. 6. Moreover, we can anticipate that large $c_H$ will lead to a rapid decay of the DM particle, and will be excluded by cosmological bounds; this will be made more precise in the next subsection. Between these two regions the production is dominated by annihilation into one DM particle.

The dependence of $T_{\text{max}}$ on $m$ and $c_H$ is different in the two regions:

$$1\text{DM }\textbf{domination}: \quad T_{\text{max}} \sim (y_{\text{ref}} M_{\text{Pl}} m^3/c_H^2)^{1/5} \sim 100\text{ GeV}\left(\frac{m}{1\text{ GeV}}\right)^{3/5}\left(\frac{1}{c_H}\right)^{2/5}, \tag{10}$$

$$2\text{DM }\textbf{domination}: \quad T_{\text{max}} \sim \left[y_{\text{ref}} M_{\text{Pl}}^3 m^5/(1+c_H^2)^2\right]^{1/9} \sim 10\text{ MeV}\left(\frac{m}{1\text{ GeV}}\right)^{5/9}\left(\frac{10^{30}}{(1+c_H^2)}\right)^{2/9}.$$

The above equation predicts $T_{\text{max}}$ smaller than the electroweak scale in a portion of the parameter space. But then, in the first place, the Higgs and top quark are not present in the heat bath to initiate the single and Higgs-mediated pair production, contrary to our initial assumptions, and therefore this region is not viable phenomenologically.

### 4.4 Quantitative results

We now calculate the DM production rate more carefully, starting from the amplitudes of Section 4.2. The amplitudes relevant for the single production process are the ones in Eqs. (5) and (7). Squaring them, integrating over the phase space, summing over all relevant degrees of freedom, inserting that into Eq. (3), and taking the limit $T \gg m$, we obtain the single production rate

$$R_{1\text{DM}} \approx \frac{12 c_H^2 T^{10}}{\pi^5 M_{\text{Pl}}^2 m^4}\left(3g^2 + g'^2 + 18y_t^2\right) \approx 0.75\,\frac{c_H^2 T^{10}}{M_{\text{Pl}}^2 m^4}, \tag{11}$$

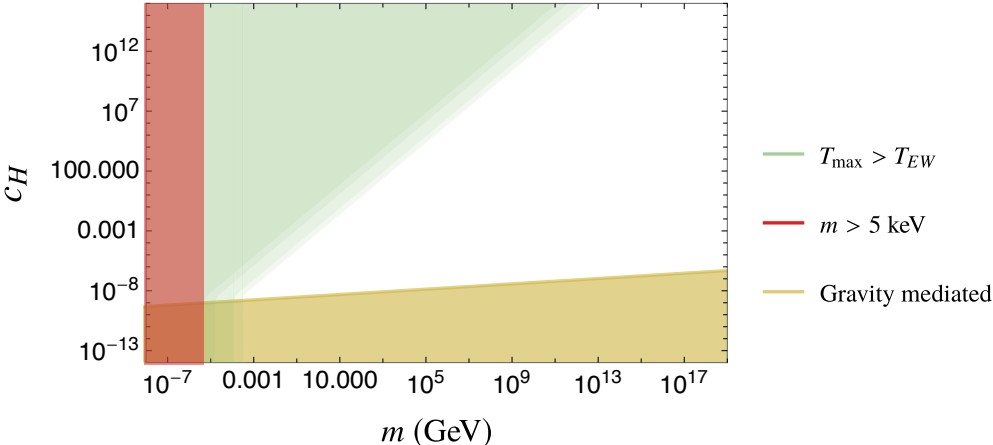

Figure 6: Spin-2 DM in the Higgs-portal scenario. We show the regions where DM production is dominated respectively by pair production with an s-channel graviton (yellow), and by single production (above the yellow region). The red region shows the small-scale structure bound $m \gtrsim 5$ keV, and in the green region the estimated $T_{\text{max}}$ is below the electroweak scale.

where we ignore the contributions proportional to the Yukawa coupling of the lighter SM fermions. The pair production rate can be split as $R_{2\text{DM}} = R_{2\text{DM}}^{(\text{GM})} + R_{2\text{DM}}^{(\text{HM})} + R_{2\text{DM}}^{(\text{int})}$ where $R_{2\text{DM}}^{(\text{GM})}$ is the gravity-mediated contribution displayed in Eq. (16), the Higgs-mediated $R_{2\text{DM}}^{(\text{HM})}$ is calculated starting from the amplitude in Eq. (4), and the $R_{2\text{DM}}^{(\text{int})}$ comes from the interference between the gravity-mediated and Higgs-mediated amplitudes. We get

$$R_{2\text{DM}}^{(\text{HM})} = \frac{11520 c_H^4 T^{14}}{\pi^5 M_{\text{Pl}}^4 m^6}, \qquad R_{2\text{DM}}^{(\text{int})} = -\frac{3840 c_H^2 T^{14}}{\pi^5 M_{\text{Pl}}^4 m^6}. \tag{12}$$

The above result is obtained without taking into account DM self-interactions, that is for $a_0 = a_2 = 0$ in Eq. (C.1); non-zero $a_0$ or $a_2$ can change the rate by an $\mathcal{O}(a_i)$ factor, but they do not affect our results in a qualitative way. Collecting the single and pair production contributions, the DM yield for $T_{\text{max}} \gg m$ is given by

$$Y_X^0 \approx 9.4 \times 10^{-4} \frac{c_H^2 T_{\text{max}}^5}{M_{\text{Pl}} m^4} + \left(0.55 - 0.0087 c_H^2 + 0.026 c_H^4\right) \frac{T_{\text{max}}^9}{M_{\text{Pl}}^3 m^6}. \tag{13}$$

For a given point in the $m$-$c_H$ parameter space, the above equation allows one to find $T_{\text{max}}$ that yields the correct relic abundance. As before, we require that $T_{\text{max}}$ is below the maximal cutoff of the EFT set by its strong coupling scale, and the $T_{\text{max}}$ is above the electroweak scale, in agreement with our initial assumptions. In this scenario there is another important constraint on the DM lifetime, since the $\mathbb{Z}_2$ breaking interaction in Eq. (1) allows the DM particle to decay. Indeed, for $m \gg 2m_Z$ we find

$$\tau_X = \frac{240\pi M_{\text{Pl}}^2}{c_H^2 m^3} \approx 3 \times 10^{26} \text{ sec} \left(\frac{10^{-10}}{c_H}\right)^2 \left(\frac{1 \text{ TeV}}{m}\right)^3. \tag{14}$$

It is clear that for heavy DM we need a tiny coupling $c_H$ to avoid the cosmological bounds on decaying dark matter. As can be seen in Fig. 6, that region corresponds to domination of gravity-mediated production, which effectively takes us back to the scenario of Section 3

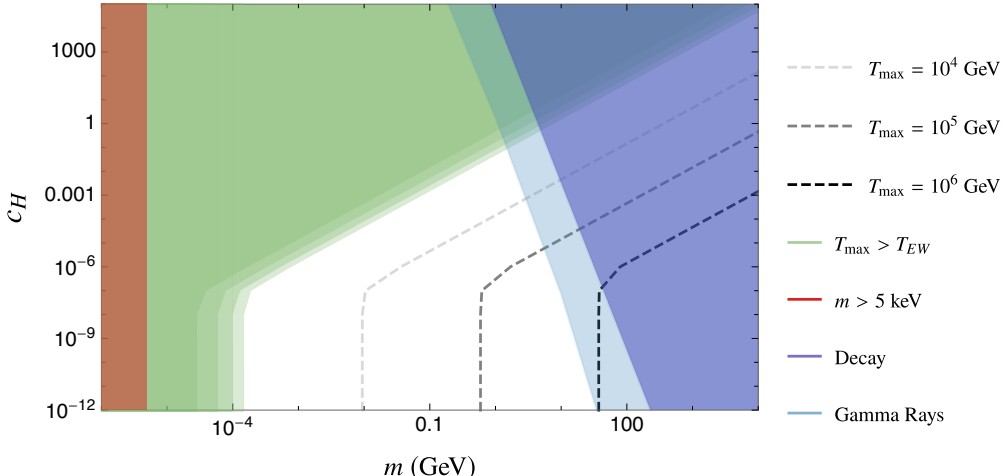

Figure 7: Spin-2 DM in the Higgs-mediated scenario. We show the contours of constant $T_{\mathrm{max}}$ leading to the correct freeze-in relic abundance. The light blue regions are excluded by the model-independent bound on the DM lifetime [52] and gamma-ray constraints [53]. The red region is excluded by the lower mass limit from small-scale structure, while in the green region ensures $T_{\mathrm{max}}$ is below the electroweak scale.

(except for the possibility of observing the DM decay). A more interesting situation occurs for smaller $m$, where 2-body decays of DM are forbidden kinematically, and instead DM decays via 3-body (for $m \sim m_Z$) and 4-body (for $m \ll m_Z$) channels. As can be observed in Fig. 7, for $m \lesssim 100$ GeV the decay constraints allow us to access the region where the single DM production dominates over the gravity-mediated pair production, leading to a scenario qualitatively different than the one in Section 3. In our analysis we used the model-independent bound on dark matter lifetime from Ref. [52], $\Gamma_X < 6.3 \times 10^{-3}$ Gyr$^{-1}$, as well as constraints from gamma-ray observations worked out in Ref. [53]. In Fig. 7 we also show the contours of $T_{\mathrm{max}}$ required to obtain the observed DM relic abundance. The characteristic "bending" of these contours occurs at the cross-over between the regions where single or pair DM production dominate the freeze-in abundance. However, in a part of the $m$-$c_H$ parameter space, $T_{\mathrm{max}}$ based on the result in Eq. (13) falls below the electroweak scale, contrary to the assumption used in that calculation. Since the top quark and the Higgs and electroweak bosons are absent from the plasma for $T < T_{\mathrm{EW}}$, the Higgs-mediated single production cannot occur, and we are again back to the gravity-mediate scenario of the previous section.[13] To summarize, we find that for $S = 2$ DM it is possible to realize the scenario of Higgs-mediated freeze-in production in a limited region of the parameter space:

$$\mathbf{S = 2}: \quad 0.1 \text{ MeV} \lesssim m \lesssim 1 \text{ TeV}, \quad 10^{-7} \lesssim |c_H| \lesssim 1, \tag{15}$$

corresponding to 1 TeV $\lesssim T_{\mathrm{max}} \lesssim 10^6$ GeV. Outside this region the parameter space is either excluded by DM decay constraints or the Higgs-mediated production is irrelevant compared to the gravity-mediated production studied in the previous section.

---

[13]In principle, DM could be single-produced from lighter fermions, if the suppression due to the small Yukawa coupling and the Boltzmann suppression is counter-balanced by very large $c_H$ (above the region plotted in Fig. 7). However, we do not study this scenario at a quantitative level in this paper.

## 4.5 Generalization to $S > 2$

The interaction in Eq. (1) can be readily generalized to arbitrary even $S$:

$$\mathcal{M}(1_{H_a} 2_{\tilde{H}_b} \, 3_X) = -\frac{c_H}{M_{\text{Pl}} m^{2S-2}} \delta_{ab} \langle \mathbf{3} p_1 \mathbf{3}]^S. \tag{16}$$

In this case the coupling scales as $c_H \sim \mathcal{O}(g_* M_{\text{Pl}} m^{S-2}/\Lambda^{S-1})$. Therefore, for $S > 2$ and $m \ll \Lambda$, one can more naturally arrive at $|c_H| \ll 1$, without invoking a very weakly coupled UV completion. In consequence, one can more naturally satisfy the DM decay constraints in the scenario with $S > 2$. For odd $S$ a similar interaction is also possible, but then Bose symmetry requires that $\delta_{ab}$ is replaced by an anti-symmetric tensor. The different DM production processes discussed earlier scale with energy as

$$\mathcal{M}_{\text{2DM}}^{(\text{GM})} \sim \frac{E^{2S+1}}{M_{\text{Pl}}^2 m^{2S-1}}, \qquad \mathcal{M}_{\text{2DM}}^{(\text{HM})} \sim \frac{c_H^2 E^{4S-3}}{M_{\text{Pl}}^2 m^{4S-5}}, \qquad \mathcal{M}_{\text{1DM}} \sim \frac{c_H E^{2S-1}}{M_{\text{Pl}} m^{2S-2}}. \tag{17}$$

Notice that only for $S = 2$, the Higgs- and gravity-mediated pair production have the same energy dependence. With the scaling in Eq. (17), the DM abundance can be estimated as

$$T_{\text{max}} \gg m \quad \rightarrow \quad Y_X^0 \sim \frac{T_{\text{max}}^{4S-3}}{M_{\text{Pl}} m^{4S-4}} \left[ c_H^2 + \frac{T_{\text{max}}^4}{M_{\text{Pl}}^2 m^2} + c_H^4 \frac{T_{\text{max}}^{4S-4}}{M_{\text{Pl}}^2 m^{4S-6}} \right]. \tag{18}$$

Single production dominates over gravitational pair production when $|c_H| > \frac{T_{\text{max}}^2}{M_{\text{Pl}} m}$. Since $T_{\text{max}}$ needed to fit the DM relic abundance decreases with increasing $S$, the region of pair production domination migrates toward lower $c_H$ as the spin is increased, as can be seen in Fig. 8. Similarly to the $S = 2$ scenario, the Higgs-mediated pair production is relevant only for very large values of $c_H$, which are not possible to obtain in an EFT with the cutoff above a TeV, and are not shown in Fig. 8. The dark matter lifetime is affected only by $\mathcal{O}(1)$ factors compared to Eq. (14). Indeed, given that $\Gamma_X$ must be proportional to $1/M_{\text{Pl}}^2$, the only available combination of dimensionful parameters with the correct dimension is $\tau_X \sim M_{\text{Pl}}^2/m^3$. Going to higher spins only leads to order one corrections with respect to Eq. (14) due to a different angular dependence of the matrix element for the decay process.

## 5 Conclusions

In this paper we described the effective field theory for a dark matter particle $X$ of arbitrary spin $S$. The on-shell amplitude techniques bring huge simplifications to the qualitative and quantitative study of higher-spin DM models. We were able to derive compact formulas for the amplitudes of SM SM $\rightarrow$ DM DM processes mediated by gravity, cf. Eqs. (3)-(5), which are valid for any $S \geq 1$. From these, the gravity-mediated pair production rate of higher-spin DM from the SM thermal bath can be readily calculated, cf. Eqs. (8) and (10). Since gravity is universal, this contribution is present in any model of higher-spin DM (though of course it can be subleading, if other stronger interactions are present). We also calculated amplitudes for processes of single and pair DM production (Eqs. (5)-(7)) that arise in the presence of the model-dependent Higgs-DM interaction in Eq. (1).

Moreover, we discussed the cosmological consequences of higher-spin DM. We focused on DM production via the freeze-in mechanism and we studied two scenarios:

1. Purely gravity-mediated pair production via minimal gravitational interactions.

2. Higgs-mediated single production in association with an electroweak gauge boson or a top quark.

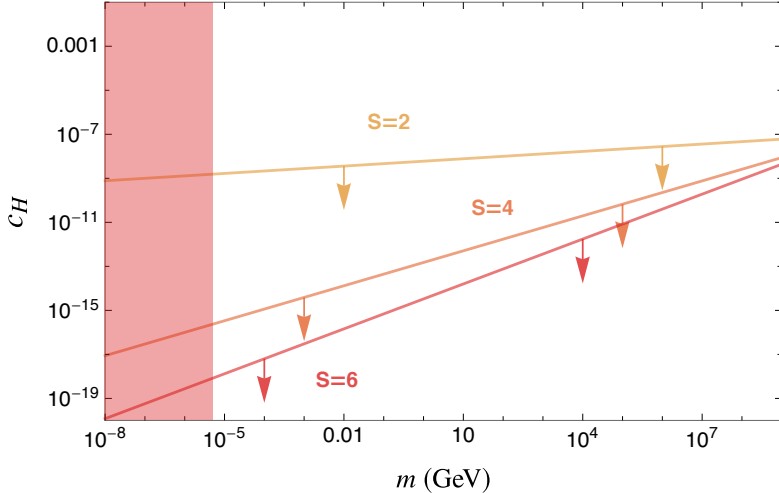

Figure 8: Comparison between spin $S = 2, 4, 6$ in the Higgs-mediation scenario. The area below each line corresponds to the region where gravity-mediated pair production dominates. In the area above the lines, single production dominates. The contributions become comparable for $T_{max} \sim \sqrt{c_H m M_{\text{Pl}}}$. The red region indicates the small-scale structure bound $m \gtrsim 5$ keV.

In the first scenario, the freeze-in DM abundance can match the experimentally observed one for a broad range of DM masses, from sub-TeV to GUT-scale masses. For longitudinal components of higher-spin particles the scale controlling the high-energy behavior of annihilation amplitudes is $[M_{\text{Pl}} m^{S-1/2}]^{1/(S+1/2)}$, which can be well below the Planck scale, especially for large $S$ and/or small $m$. Therefore, the minimal gravitational interactions alone can be strong enough to allow for sufficient DM production (and simultaneously remain in the freeze-in regime). The challenge for this scenario is to avoid that other processes, such as Compton and self-scattering, hit the strong coupling before annihilation becomes strong enough. This can happen for smaller DM masses, and the constraint becomes more restrictive as $S$ is increased. Nevertheless, even for large $S$ there remains a window to generate correct abundance of superheavy DM right at the production threshold where DM is produced with non-relativistic velocities.

The gravity-mediated scenario in its minimal form does not have any experimental signatures, other than the usual gravitational effects of DM on the dynamics of the Universe. The situation is phenomenologically more interesting in the second scenario. Given that the cubic DM-Higgs interaction we introduced violates the $\mathbb{Z}_2$ symmetry, the DM particle is no longer stable, leading to potentially rich signatures of decaying DM. In fact, the decays provide stringent constraint on the parameter space, and exclude sizable DM-Higgs couplings, unless the DM particle is much lighter than the electroweak scale. In general, these constraints push the Higgs-mediated scenario toward the parameter region where the pure gravity-mediated processes dominate DM production. Nevertheless, we do find viable parameter regions where the Higgs-mediated single production dominates.

We comment here on the relationship between our work and recent Refs. [36, 37], which also considered higher-spin DM. One feature that sets apart our paper is that we apply the massive spinor formalism to calculate the DM amplitudes. This is a technical point, but a very important one, as it allows us to obtain compact analytic formulas for the amplitudes and study a broad range of possible interactions. In particular, it allows us to tackle gravity mediated amplitudes, which may be challenging in other formulations, and was not attempted in [36,

37].[14] Our work also differs from Refs. [36, 37] regarding the details of the model and the parametric regime. In Ref. [36], DM interacts with the SM via 4-point Higgs-portal amplitudes. This interaction corresponds to the contact term $C_\phi^{(S)}$ in Eq. (3), but in our scenario we assume it is subleading for the sake of freeze-in production (in Section 3.5 we briefly discussed a scenario were a different contact term dominates over gravitational interactions, which leads to similar physics). Thus, Ref. [36] corresponds to the parametric limit of our model where the contact term $C_\phi^{(S)}$ dominates. Ref. [37] studies gravitational DM production, and it employs the same minimal coupling to gravity (since it is universal). However, DM production occurs in the de Sitter phase *during* inflation, while we are considering the production during the reheating phase *after* inflation ends. The two mechanisms are complementary and the question which one dominates is a model-dependent one, depending on the relationship between the DM mass, reheating temperature, and the Hubble constant during inflation; e.g. the mechanism of Ref. [37] is not efficient if $m_{\mathrm{DM}}$ is below $H$. Note that our higher-spin DM particle is not elementary, and likely corresponds to some bound state, thus it may not even exist during inflation.

There are countless ways we could deform or generalize the spin-$S$ DM model studied in this paper. Instead through the Higgs, spin-$S$ DM could couple to the SM via vector or fermion portals. Moreover, it is unlikely a higher-spin state exists in isolation. Indeed, concrete frameworks like large $N$ Yang-Mills or string theory lead to families or towers of states with varying spins. It would be interesting to study DM in higher-spin EFTs derived from realistic UV completions [19, 54, 55]. In all these investigations, we expect, the on-shell methods will play a crucial role, as they facilitate the calculations and provide better physical insight.

## Acknowledgments

We thank Marco Cirelli and Yann Mambrini for useful discussions and suggestions. AF is partially supported by the Agence Nationale de la Recherche (ANR) under grant ANR-19-CE31-0012 (project MORA).

## A    Spinor conventions

We use the mostly-minus metric signature: $\eta_{\mu\nu} = \mathrm{diag}(1, -1, -1, -1)$. The basic objects from which we build scattering amplitudes are 2-component holomorphic and anti-holomorphic spinors, transforming respectively in $(1/2, 0)$ and $(0, 1/2)$ representations of the Lorentz group. Following the conventions of Ref. [56], the spinor indices are lowered and raised by the antisymmetric $\epsilon$ tensor:

$$\psi^\alpha = \epsilon^{\alpha\beta}\psi_\beta \,, \qquad \psi_\alpha = \epsilon_{\alpha\beta}\psi^\beta \,, \tag{A.1}$$

with $\epsilon^{12} = -\epsilon_{12} = 1$. Vector and spinor Lorentz indices can be traded via the sigma matrices $[\sigma^\mu]_{\alpha\dot\beta} = (1, \vec\sigma)$, $[\bar\sigma^\mu]^{\dot\alpha\beta} = (1, -\vec\sigma)$, where $\vec\sigma = (\sigma^1, \sigma^2, \sigma^3)$ are the usual Pauli matrices. Massless momenta $p^\mu$ can be represented by two spinors denoted as $\lambda_\alpha$, $\tilde\lambda_{\dot\beta}$. They are related to the momentum via the equation

$$(p_i \sigma)_{\alpha\dot\beta} = \lambda_{i\,\alpha}\tilde\lambda_{i\,\dot\beta}, \qquad (p\bar\sigma)^{\dot\alpha\beta} = \tilde\lambda_i^{\dot\alpha}\lambda_i^\beta \,, \tag{A.2}$$

such that $p^2 = 0$. The massless *little group* of the $i$-th particle corresponds to $U(1)$ acting as $\lambda_i \to t_i^{-1}\lambda_i$, $\tilde\lambda_i \to t_i\tilde\lambda_i$. An amplitude with the $i$-th particle having helicity $h_i$ must transform

---

[14]Note that gravity mediated amplitudes are universal and our results are applicable beyond the specific field of DM, for example for calculating quantum corrections to classical gravitational interactions of higher-spin objects.

as $\mathcal{M} \to t_i^{2h_i} \mathcal{M}$ under this $U(1)$. For each massive momentum we have four associated spinors denoted as $\chi_\alpha^1, \chi_\alpha^2, \tilde{\chi}_{\dot{\beta}\,1}, \tilde{\chi}_{\dot{\beta}\,2}$. They are related to the momentum via [15]

$$(p_i \sigma)_{\alpha\dot{\beta}} = \sum_{J=1}^{2} \chi_{i\,\alpha}^{J} \tilde{\chi}_{i\,\dot{\beta}\,J}, \qquad (p_i \bar{\sigma})^{\dot{\alpha}\beta} = \sum_{J=1}^{2} \tilde{\chi}_{i\,J}^{\dot{\alpha}} \chi_i^{\beta\,J}, \tag{A.3}$$

and subject to the normalization conditions

$$\chi^J \chi_K = \delta_K^J m, \qquad \tilde{\chi}_J \tilde{\chi}^K = \delta_J^K m. \tag{A.4}$$

The index $J$ is associated with the $SU(2)$ little group of a massive particle. An amplitude with the $i$-th particle having spin $S$ must be a sum of analytic functions of exactly $2S$ spinors $\chi_i$ or $\tilde{\chi}_i$.

Lorentz-invariant spinor contractions are abbreviated using the bra-ket notation. For massless spinors

$$\langle ij \rangle \equiv \lambda_i^\alpha \lambda_{j\,\alpha} = \epsilon^{\beta\alpha} \lambda_{i\,\alpha} \lambda_{j\,\beta} = (\lambda_i \lambda_j), \qquad [ij] \equiv \tilde{\lambda}_{i\,\dot{\alpha}} \tilde{\lambda}_j^{\dot{\alpha}} = \epsilon^{\dot{\alpha}\dot{\beta}} \tilde{\lambda}_{i\,\dot{\alpha}} \lambda_{j\,\dot{\beta}} = (\tilde{\lambda}_i \tilde{\lambda}_j), \tag{A.5}$$

and momentum insertions are represented by $(\lambda_i p_k \sigma \tilde{\lambda}_j) \equiv \langle i p_k j], (\lambda_i p_k \sigma p_l \bar{\sigma} \tilde{\lambda}_j) \equiv \langle i p_k p_l j]$, etc. Similarly, for massive spinors

$$\langle \mathbf{ij} \rangle \equiv \chi_i^\alpha \chi_{j\,\alpha} = (\chi_i \chi_j), \qquad [\mathbf{ij}] \equiv \tilde{\chi}_{i\,\dot{\alpha}} \tilde{\chi}_j^{\dot{\alpha}} = (\tilde{\chi}_i \tilde{\chi}_j), \tag{A.6}$$

with the only difference that **bold** notation is used in this case.

For a massive particle with the momentum $p^\mu = (\sqrt{|p|^2 + m^2}, |p|\sin\theta\cos\phi, |p|\sin\theta\sin\phi, |p|\cos\theta)$ an explicit form of the associated spinors is

$$
\chi_\alpha^J = \begin{pmatrix} \sqrt{E-|p|}\cos\frac{\theta}{2} & -\sqrt{E+|p|}e^{-i\phi}\sin\frac{\theta}{2} \\ \sqrt{E-|p|}e^{i\phi}\sin\frac{\theta}{2} & \sqrt{E+|p|}\cos\frac{\theta}{2} \end{pmatrix},
$$
$$
\tilde{\chi}_{\dot{\alpha}}^J = \begin{pmatrix} -\sqrt{E+|p|}e^{i\phi}\sin\frac{\theta}{2} & -\sqrt{E-|p|}\cos\frac{\theta}{2} \\ \sqrt{E+|p|}\cos\frac{\theta}{2} & -\sqrt{E-|p|}e^{-i\phi}\sin\frac{\theta}{2} \end{pmatrix}. \tag{A.7}
$$

Of course, any $SU(2)$ rotation of the above also gives a valid spinor decomposition of a massive momentum.

# B  Derivation of production amplitudes

## B.1  Gravity-mediated pair production

We consider the 4-point amplitude $\mathcal{M}(1_f 2_f \mathbf{3}_X \mathbf{4}_X)$ where $f$ is a massless particle of helicity $h = 0, \pm 1/2, \pm 1$, and $X$ is a massive particle of arbitrary spin $S$. In any consistent theory containing gravity this amplitude has a pole in the s-channel corresponding to a massless graviton exchange. The residue of that pole is given by

$$R_s = -\sum_{h'\pm} \mathcal{M}(1_f 2_f (-p_s)^{h'}) \mathcal{M}((p_s)^{-h'} \mathbf{3}_X \mathbf{4}_X)|_{s\to 0}, \tag{B.1}$$

where $p_s = p_1 + p_2$, and the sum goes over the two helicities $h$ of the intermediate graviton. Assuming minimal coupling to gravity, the relevant 3-point amplitudes to compute the residue are given in Eq. (1) and Eq. (2). We get

$$R_s = -\frac{1}{M_{\text{Pl}}^2 m^{2S}} \left\{ (A_{12}^+)^{2-2|h|}(B_{12}^+)^{2|h|}(A_{34}^-)^2 [\mathbf{43}]^{2S} + (A_{12}^-)^{2-2|h|}(B_{12}^-)^{2|h|}(A_{34}^+)^2 \langle \mathbf{43} \rangle^{2S} \right\}|_{s\to 0}, \tag{B.2}$$

where $A_{12}^+ = -\langle\zeta p_1 p_s]/\langle p_s\zeta\rangle, A_{12}^- = -[\zeta p_1 p_s\rangle/[p_s\zeta], A_{34}^+ = \langle\zeta p_3 p_s]/\langle p_s\zeta\rangle, A_{34}^- = [\zeta p_3 p_s\rangle/[p_s\zeta],$
$B_{12}^+ = \langle 1\zeta\rangle[2p_s]/\langle p_s\zeta\rangle$ and $B_{12}^- = \langle 1p_s\rangle[2\zeta]/[p_s\zeta]$, with $|\zeta\rangle$ and $|\zeta]$ arbitrary reference spinors orthogonal to $p_s$. We can trivially rewrite it as

$$
\begin{aligned}
R_s = & -\frac{1}{2M_{\text{Pl}}^2 m^{2S}}\Bigg\{\Big((A_{12}^+)^{2-2|h|}(B_{12}^+)^{2|h|}(A_{34}^-)^2 + (A_{12}^-)^{2-2|h|}(B_{12}^-)^{2|h|}(A_{34}^+)^2\Big)\Big([43]^{2S} + \langle 43\rangle^{2S}\Big) \\
& + \Big((A_{12}^+)^{2-2|h|}(B_{12}^+)^{2|h|}(A_{34}^-)^2 - (A_{12}^-)^{2-2|h|}(B_{12}^-)^{2|h|}(A_{34}^+)^2\Big)\Big([43]^{2S} - \langle 43\rangle^{2S}\Big)\Bigg\}\Big|_{s\to 0}.
\end{aligned} \tag{B.3}
$$

Now, we can prove the identities that hold in the limit $s\to 0$:

$$
\begin{aligned}
A_{12}^+ A_{34}^- + A_{12}^- A_{34}^+ &= -2p_1 p_3, \\
B_{12}^+ A_{34}^- + B_{12}^- A_{34}^+ &= -\langle 1p_3 2],
\end{aligned} \tag{B.4}
$$

$$
\begin{aligned}
\big(A_{12}^+ A_{34}^- - A_{12}^- A_{34}^+\big)\big(\langle 34\rangle - [34]\big) &= -2p_1 p_3\big(\langle 34\rangle + [34]\big) - 2m\big(\langle 3p_1 4] + \langle 4p_1 3]\big), \\
\big(B_{12}^+ A_{34}^- - B_{12}^- A_{34}^+\big)\big(\langle 34\rangle - [34]\big) &= -\langle 1p_3 2]\big(\langle 34\rangle + [34]\big) + 2m\big(\langle 31\rangle[42] + \langle 41\rangle[12]\big).
\end{aligned} \tag{B.5}
$$

This together with

$$
A_{12}^+ A_{12}^- = 0, \quad A_{34}^- A_{34}^+ = m^2, \quad B_{12}^+ B_{12}^- = 0, \tag{B.6}
$$

leads to

$$
(A_{12}^+)^{2-2|h|}(B_{12}^+)^{2|h|}(A_{34}^-)^2 + (A_{12}^-)^{2-2|h|}(B_{12}^-)^{2|h|}(A_{34}^+)^2 = (2p_1 p_3)^{2-2|h|}\langle 1p_3 2]^{2|h|} \tag{B.7}
$$

$$
\begin{aligned}
\Big((A_{12}^+)^{2-2|h|}(B_{12}^+)^{2|h|}(A_{34}^-)^2 &- (A_{12}^-)^{2-2|h|}(B_{12}^-)^{2|h|}(A_{34}^+)^2\Big)\big(\langle 34\rangle - [34]\big) \\
&= (2p_1 p_3)^{2-|2h|}\langle 1p_3 2]^{|2h|}\big(\langle 34\rangle + [34]\big) + 2mF_{|h|},
\end{aligned} \tag{B.8}
$$

where $F_0 \equiv 2p_1 p_3\big(\langle 3p_1 4] + \langle 4p_1 3]\big), F_{1/2} \equiv -2p_1 p_3\big(\langle 31\rangle[42] + \langle 41\rangle[32]\big),$
$F_1 \equiv -\langle 1p_3 2]\big(\langle 31\rangle[42] + \langle 41\rangle[32]\big)$. Armed with these formulae, we can write down the $s$-residue as

$$
\begin{aligned}
R_s = -\frac{1}{2M_{\text{Pl}}^2 m^{2S}}\Bigg\{&(2p_1 p_3)^{2-2|h|}\langle 1p_3 2]^{2|h|}\Big([43]^{2S} + \langle 43\rangle^{2S}\Big) \\
&+ \Big((2p_1 p_3)^{2-|2h|}\langle 1p_3 2]^{|2h|}\big(\langle 34\rangle + [34]\big) + 2mF_{|h|}\Big)\sum_{k=0}^{2S-1}[43]^k\langle 43\rangle^{2S-1-k}\Bigg\}\Big|_{s\to 0},
\end{aligned} \tag{B.9}
$$

or, simplifying,

$$
R_s = \frac{1}{M_{\text{Pl}}^2 m^{2S}}\Bigg\{(2p_1 p_3)^{2-|2h|}\langle 1p_3 2]^{|2h|}\sum_{k=1}^{2S-1}[43]^k\langle 43\rangle^{2S-k} - mF_{|h|}\sum_{k=0}^{2S-1}[43]^k\langle 43\rangle^{2S-1-k}\Bigg\}\Big|_{s\to 0}. \tag{B.10}
$$

We can trade $2p_1 p_3$ for $(t-u)/2$ on the $s$-pole. The last step is to isolate the term proportional to $s$ from the first term in the curly bracket. For example, for $h = 1$ this can be by applying the identities

$$
\begin{aligned}
\langle 1p_3 2]\langle 43\rangle &= [12]\langle 31\rangle\langle 41\rangle - m\big(\langle 31\rangle[42] + \langle 41\rangle[32]\big), \\
\langle 1p_3 2][43] &= -\langle 12\rangle[32][42] - m\big(\langle 31\rangle[42] + \langle 41\rangle[32]\big).
\end{aligned} \tag{B.11}
$$

This allows one to transform Eq. (B.10) into

$$
\begin{aligned}
R_s &= \frac{\langle 31\rangle[42]+\langle 41\rangle[32]}{M_{\text{Pl}}^2 m^{2S-1}}\Bigg\{\langle 1p_3 2]\sum_{k=0}^{2S-1}[43]^k\langle 43\rangle^{2S-1-k}\\
&\quad+\Big(\langle 12\rangle[32][42]-[12]\langle 31\rangle\langle 41\rangle+m\big(\langle 31\rangle[42]+\langle 41\rangle[32]\big)\Big)\sum_{k=1}^{2S-1}[43]^{k-1}\langle 43\rangle^{2S-k-1}\Bigg\}\Big|_{s\to 0}.
\end{aligned}
$$
(B.12)

This has clearly the same UV behavior as the pole term in Eq. (5). Applying in addition the identity

$$
\langle 12\rangle[32][42]-[12]\langle 31\rangle\langle 41\rangle=-2m\big(\langle 31\rangle[42]+\langle 41\rangle[32]\big)-\langle 1p_3 2]\big(\langle 43\rangle+[43]\big),\quad\text{(B.13)}
$$

one recovers precisely the s-pole residue in Eq. (5). For $h=0,1/2$, similar steps lead from Eq. (B.10) to Eqs. (3) and (4).

## B.2 Higgs-mediated pair production

We consider the amplitude $\mathcal{M}(1_H 2_H 3_X 4_X)$ where $H$ is a spin-0 particle and $X$ is a spin-2 particle. The two interact via the 3-point amplitude in Eq. (1). From that, the $t$- and $u$-channel residues are simply obtained:

$$
R_t=-\frac{c_H^2}{M_{\text{Pl}}^2 m^4}\langle 3p_1 3]^2\langle 4p_2 4]^2|_{t\to 0},\qquad R_u=-\frac{c_H^2}{M_{\text{Pl}}^2 m^4}\langle 3p_2 3]^2\langle 4p_1 4]^2|_{u\to 0}.
$$
(B.14)

We then use the identities

$$
\begin{aligned}
\langle 3p_1 3]\langle 4p_2 4] &= (t-m^2)\langle 34\rangle[34]+m\big(\langle 34\rangle\langle 3p_1 4]+[34]\langle 4p_1 3]\big)-\langle 3p_1 4]\langle 4p_1 3],\\
\langle 3p_2 3]\langle 4p_1 4] &= (u-m^2)\langle 34\rangle[34]-m\big(\langle 34\rangle\langle 4p_1 3]+[34]\langle 3p_1 4]\big)-\langle 3p_1 4]\langle 4p_1 3],
\end{aligned}\quad\text{(B.15)}
$$

and also

$$
\langle 3p_1 3]\langle 4p_2 4]u+\langle 3p_2 3]\langle 4p_1 4]t=(tu-m^4)\langle 34\rangle[34]-m^2\big(\langle 3p_1 4]\langle 3p_2 4]+\langle 4p_1 3]\langle 4p_2 3]\big),\quad\text{(B.16)}
$$

to rewrite the residues as

$$
\begin{aligned}
R_t &= -\frac{c_H^2}{M_{\text{Pl}}^2 m^3}\Bigg\{\langle 3p_1 3]\langle 4p_2 4]\Big(\langle 34\rangle\langle 3p_1 4]+[34]\langle 4p_1 3]-m\langle 34\rangle[34]\Big)\\
&\quad+\frac{m\langle 3p_1 4]\langle 4p_1 3]}{u}\Big(\langle 3p_1 4]\langle 3p_2 4]+\langle 4p_1 3]\langle 4p_2 3]+m^2\langle 34\rangle[34]\Big)\Bigg\}\Big|_{t\to 0},\\
R_u &= -\frac{c_H^2}{M_{\text{Pl}}^2 m^3}\Bigg\{-\langle 3p_2 3]\langle 4p_1 4]\Big(\langle 34\rangle\langle 4p_1 3]+[34]\langle 3p_1 4]+m\langle 34\rangle[34]\Big)\\
&\quad+\frac{m\langle 3p_1 4]\langle 4p_1 3]}{t}\Big(\langle 3p_1 4]\langle 3p_2 4]+\langle 4p_1 3]\langle 4p_2 3]+m^2\langle 34\rangle[34]\Big)\Bigg\}\Big|_{u\to 0}.
\end{aligned}
$$
(B.17)

Up to contact terms, the full amplitude can be reconstructed as

$$
\begin{aligned}
\mathcal{M}(1_H 2_H 3_X 4_X) &= -\frac{c_H^2}{M_{\text{Pl}}^2 m^3}\Bigg\{\frac{\langle 3p_1 3]\langle 4p_2 4]}{t}\Big(\langle 34\rangle\langle 3p_1 4]+[34]\langle 4p_1 3]-m\langle 34\rangle[34]\Big)\\
&\quad-\frac{\langle 3p_2 3]\langle 4p_1 4]}{u}\Big(\langle 34\rangle\langle 4p_1 3]+[34]\langle 3p_1 4]+m\langle 34\rangle[34]\Big)\\
&\quad+\frac{m\langle 3p_1 4]\langle 4p_1 3]}{tu}\Big(\langle 3p_1 4]\langle 3p_2 4]+\langle 4p_1 3]\langle 4p_2 3]+m^2\langle 34\rangle[34]\Big)\Bigg\}.
\end{aligned}
$$
(B.18)

Dropping the sub-leading terms in $1/m$, one recovers the high-energy limit of this amplitude displayed in Eq. (4).

### B.3 Single production

We compute the amplitude $\mathcal{M}(1_H 2_{\bar{H}} 3_X 4_v)$, where $H$ is the Higgs doublet and $v$ is an electroweak vector boson in the SM. By crossing, this amplitude encodes information about the $HH^\dagger \to Xv$ and $Hv \to XH$ processes, relevant for single DM freeze-in production. We only show a derivation for negative helicity $v$; for positive helicity all steps are analogous. The amplitude has poles in the $t$- and $u$-channels, corresponding to the Higgs exchange. Given the 3-point amplitudes in Eq. (1) and (2) the residues of these poles can be calculated as

$$R_t = -\mathcal{M}(1_{H_a}(-p_t)_{H_e} 3_X)\mathcal{M}((p_t)_{H_e} 2_{H_b} 4_{v_c}^-)|_{t\to 0} = \frac{c_H\sqrt{2}g_v T_{ab}^c}{M_{\text{Pl}} m^2 u}\langle 4p_1 p_2 4\rangle[3p_1 3]^2|_{t\to 0},$$

$$R_u = -\mathcal{M}(1_{H_a}(-p_u)_{H_e} 4_{v_c}^-)\mathcal{M}((p_u)_{H_e} 2_{H_b} 3_X)|_{u\to 0} = \frac{c_H\sqrt{2}g_v T_{ab}^c}{M_{\text{Pl}} m^2 t}\langle 4p_1 p_2 4\rangle[3p_2 3]^2|_{u\to 0}. \quad \text{(B.19)}$$

In this case the residue in the t-channel contains a pole in the u-channel, and vice-versa, therefore some work is needed to reconstruct a valid amplitude consistent with unitarity and locality. To this end we first rearrange the residues using the identities

$$[3p_1 3]^2 = \frac{1}{2}\left([3p_1 3]^2 + [3p_2 3]^2 + [3p_4 3]^2\right) + \langle 3p_4 3][3p_2 3],$$

$$[3p_2 3]^2 = \frac{1}{2}\left([3p_1 3]^2 + [3p_2 3]^2 + [3p_4 3]^2\right) + \langle 3p_4 3][3p_1 3]. \quad \text{(B.20)}$$

Next, we use the Fierz decomposition

$$\langle 3p_2 3][4p_1 p_2 4\rangle = -u\langle 43\rangle[4p_2 3] - m\langle 4p_1 3][4p_2 3],$$

$$\langle 3p_1 3][4p_1 p_2 4\rangle = t\langle 43\rangle[4p_1 3] + m\langle 4p_1 3][4p_2 3]. \quad \text{(B.21)}$$

Finally, we apply the identity

$$[4p_1 3][4p_2 3][3p_4 3] = \frac{\langle 43\rangle}{m}\left(u[4p_2 3][3p_1 3] + t[4p_1 3][3p_2 3]\right). \quad \text{(B.22)}$$

This allows us to rewrite the residues as

$$R_t = \frac{c_H\sqrt{2}g_v T_{ab}^c}{M_{\text{Pl}} m^2}\left\{\frac{\langle 4p_1 p_2 4\rangle}{2u}\left([3p_1 3]^2 + [3p_2 3]^2 + [3p_4 3]^2\right) + \langle 43\rangle[4p_2 3][3p_2 3]\right\}\Big|_{t\to 0},$$

$$R_u = \frac{c_H\sqrt{2}g_v T_{ab}^c}{M_{\text{Pl}} m^2}\left\{\frac{\langle 4p_1 p_2 4\rangle}{2t}\left([3p_1 3]^2 + [3p_2 3]^2 + [3p_4 3]^2\right) - \langle 43\rangle[4p_1 3][3p_1 3]\right\}\Big|_{u\to 0}. \quad \text{(B.23)}$$

Thanks to this massaging, the parts of the residues containing the pole are symmetric between $t$- and $u$-channels. We are ready to reconstruct the full amplitude:

$$\mathcal{M}(1_{H_a} 2_{\bar{H}_b} 3_X 4_{v_c}^-) = \frac{c_H\sqrt{2}g T_{ab}^c}{M_{\text{Pl}} m^2}\left\{\frac{\langle 4p_1 p_2 4\rangle}{2tu}\left([3p_1 3]^2 + [3p_2 3]^2 + [3p_4 3]^2\right)\right.$$

$$\left. + \langle 43\rangle\left(\frac{[4p_2 3][3p_2 3]}{t} - \frac{[4p_1 3][3p_1 3]}{u}\right)\right\}, \quad \text{(B.24)}$$

up to possible contact terms.

These results can be easily generalized to a DM with arbitrary even spin. In this case, the cubic DM-Higgs coupling is given by Eq. (16), which allows us to write the residues in the $t$ and $u$ poles as

$$R_t = \frac{c_H\sqrt{2}g_v T_{ab}^c}{M_{\text{Pl}} m^{2S-2} u}\langle 4p_1 p_2 4\rangle[3p_1 3]^S|_{t\to 0}, \qquad \frac{c_H\sqrt{2}g_v T_{ab}^c}{M_{\text{Pl}} m^{2S-2} t}\langle 4p_1 p_2 4\rangle[3p_2 3]^S|_{u\to 0}. \quad \text{(B.25)}$$

Just using momentum conservation and the binomial expansion we can generalize Eq. (B.20) to

$$
\langle \mathbf{3} p_1 \mathbf{3}]^S = \frac{1}{2} \left( [\mathbf{3} p_1 \mathbf{3}\rangle^S + [\mathbf{3} p_2 \mathbf{3}\rangle^S + [\mathbf{3} p_4 \mathbf{3}\rangle^S \right) + \frac{1}{2} \langle \mathbf{3} p_4 \mathbf{3}] \langle \mathbf{3} p_2 \mathbf{3}] \sum_{k=1}^{S-1} \binom{S}{k} \langle \mathbf{3} p_2 \mathbf{3}]^{S-k-1} \langle \mathbf{3} p_4 \mathbf{3}]^{k-1},
$$

$$
\langle \mathbf{3} p_2 \mathbf{3}]^S = \frac{1}{2} \left( [\mathbf{3} p_1 \mathbf{3}\rangle^S + [\mathbf{3} p_2 \mathbf{3}\rangle^S + [\mathbf{3} p_4 \mathbf{3}\rangle^S \right) + \frac{1}{2} \langle \mathbf{3} p_4 \mathbf{3}] \langle \mathbf{3} p_1 \mathbf{3}] \sum_{k=1}^{S-1} \binom{S}{k} \langle \mathbf{3} p_1 \mathbf{3}]^{S-k-1} \langle \mathbf{3} p_4 \mathbf{3}]^{k-1},
$$

$$(B.26)$$

and apply the identities shown in Eqs. (B.21)-(B.22). We can then reconstruct the full amplitude for the single production of an even spin-$S$ DM:

$$
\mathcal{M}(1_{H_a} 2_{\bar{H}_b} 3_X 4_{\nu_c}^-) = \frac{c_H g T_{ab}^c}{\sqrt{2}\, M_{\mathrm{Pl}} m^{2S-2}} \Bigg\{ \frac{\langle 4 p_1 p_2 4 \rangle}{tu} \left( [\mathbf{3} p_1 \mathbf{3}\rangle^S + [\mathbf{3} p_2 \mathbf{3}\rangle^S + [\mathbf{3} p_4 \mathbf{3}\rangle^S \right)
$$
$$
+ \langle 43 \rangle \sum_{k=1}^{S-1} \binom{S}{k} \langle \mathbf{3} p_4 \mathbf{3}]^{k-1} \left( \frac{\langle 4 p_2 \mathbf{3}] \langle \mathbf{3} p_2 \mathbf{3}]^{S-k}}{t} - \frac{\langle 4 p_1 \mathbf{3}] \langle \mathbf{3} p_1 \mathbf{3}]^{S-k}}{u} \right) \Bigg\}.
$$

$$(B.27)$$

## C  DM self-interaction

For a spin-2 DM particle, once the $\mathbb{Z}_2$ symmetry is broken by Eq. (1), nothing forbids a cubic self-interaction. Its presence affects the amplitude for Higgs annihilating into DM, therefore we quote it here for completeness. We assume the 3-point self-interaction amplitude of the form

$$
\mathcal{M}(1_X 2_X 3_X) = -c_H \frac{[\mathbf{3}\sigma^\mu \mathbf{3}\rangle [\mathbf{3}\sigma^\nu \mathbf{3}\rangle}{4 M_{\mathrm{Pl}} m^6} \Big[ a_0 m^2 \langle \mathbf{12}\rangle [\mathbf{12}] \langle \mathbf{1}\sigma^\mu \mathbf{1}\rangle \langle \mathbf{2}\sigma^\nu \mathbf{2}] \tag{C.1}
$$
$$
+ a_2 \big( \langle \mathbf{12}\rangle^2 p_{12}^\mu - (\langle \mathbf{12}\rangle + [\mathbf{12}]) Y^\mu \big) \big( [\mathbf{12}]^2 p_{12}^\nu - (\langle \mathbf{12}\rangle + [\mathbf{12}]) \tilde{Y}^\nu \big) \Big],
$$

where $a_0$ and $a_2$ are numerical parameters, $Y^\mu \equiv [\mathbf{1}\sigma^\mu \mathbf{2}\rangle + [\mathbf{2}\sigma^\mu \mathbf{1}\rangle + p_{12}^\mu \langle \mathbf{12}\rangle$, $\tilde{Y}^\mu \equiv [\mathbf{1}\sigma^\mu \mathbf{2}\rangle + [\mathbf{2}\sigma^\mu \mathbf{1}\rangle + p_{12}^\mu [\mathbf{12}]$ and $p_{12}^\mu \equiv p_1^\mu - p_2^\mu$. Eq. (C.1) is up to normalization identical to the spin-2 massive graviton self-interaction in the DRGT gravity [50], and leads to scattering amplitude that are maximally well-behaved in the high-energy limit [51].

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
