# Peer review of "On-shell effective theory for higher-spin dark matter"

_SciPost Physics, doi:SciPost Phys. 10, 101 (2021)_

## Round 2 · Referee Report · Anonymous · 2021-1-11

Report

This paper presents an effective field theory for a dark matter particle of arbitrary spin, using on-shell amplitude techniques. DM is produced via the freeze-in mechanism, motivated by perturbativity of the (irrelevant) interactions giving rise to the correct abundance. The preferred range of DM masses is computed and the challenges of this scenario, in particular concerning the validity of the EFT, are addressed.

It is my opinion that this is a novel and interesting work, the study is scientifically sound and detailed, and the manuscript is well-written and clear. Overall, I believe it meets the standards for publication in SciPost.

There are only several minor points that perhaps the authors could comment on:

- What is the expected size of the coefficients associated to the contact terms? How do they scale in terms of coupling constants ($\hbar$ counting)? In this regard, does it make sense to consider $C_v^{(2)} \gg 1$ in Figure 4? A comment could be explicitly made (if it is not there already, apologies if I have missed it), that the overall normalization by $M_{Pl}^2$ could be counterbalanced by a large $C_v^{(2)} \sim (M_{Pl}/\Lambda)^2$. Same question/comment goes for $c_H$.

- Why does the estimate of $T_{max}$ in Eq. (4.11) differs so much from the actual computation in Eq. (4.14)? This in turn makes the green region corresponding to $T_{max} > T_{EW}$ differ significantly from Figure 6 to Figure 7. Also, which of the two estimates in Eq. (4.11) is used for Figure 6?

- In the case that the DM candidate is not stable, is there any advantage from $X$ having a high spin?
I believe, from simple dimensional analysis, that the lifetime will not depend on the spin, although it could be nice to explicitly comment on this point.

- Are the authors aware of explicit UV completions where the lightest stable particle is of high spin? Any idea why this could be the case?

  • validity: top
  • significance: high
  • originality: high
  • clarity: high
  • formatting: perfect
  • grammar: -

Author:  Giulia Isabella  on 2021-02-16  [id 1246]

(in reply to Report 1 on 2021-01-11)
Category:
answer to question
correction

We thank the referee for the helpful comments and all the interesting points raised. Below, we address the referee's points sequentially, and detail the changes we have made in the draft to address their helpful feedback. For convenience we list the questions and respective answers below.

"What is the expected size of the coefficients associated to the contact terms? How do they scale in terms of coupling constants ($\hbar$ counting)? In this regard, does it make sense to consider $C_v^{(2)} \gg 1$ in Figure 4? A comment could be explicitly made (if it is not there already, apologies if I have missed it), that the overall normalization by $ M_{\rm Pl}^2$ could be counterbalanced by a large $C_v^{(2)} \sim ( M_{\rm Pl}/\Lambda)^2$. Same question/comment goes for $c_H$."

Below Eq.~(3.6) we added a short discussion about the scaling of the Wilson coefficients and the $\hbar$ counting. Indeed the Wilson coefficients are $\mathcal{O}(g_*^2 M_{\rm Pl}^2/\Lambda^2)$, so coefficients much larger than 1 are consistent with the effective theory assuming new physics producing these contact terms enters below the Planck scale.
We also added a comment on the scaling of $c_H$ below Eqs.~(4.1) and (4.16).

"Why does the estimate of $T_{\rm max}$ in Eq. (4.11) differs so much from the actual computation in Eq. (4.14)? This in turn makes the green region corresponding to $T_{\rm max}>T_{\rm EW}$ differ significantly from Figure 6 to Figure 7. Also, which of the two estimates in Eq. (4.11) is used for Figure 6?"

We thank the referee for this comment. Both Eq.~(4.11) (4.10 in the new version) and Eq.~(4.14) (now 4.13) are correct, however a coding mistake has been made when plotting the green region on Figure 6. The figure has been replaced, and now the green region matches the corresponding region in Figure 7.
When addressing this comment we also spotted a typo in the code related to the position of the violet region in Fig.~6 (Higgs-mediated pair production). After correcting this mistake, the violet region shifted to huge valus of $c_H$, which are not relevant withing the validity of the EFT. For this reason do not plot anymore the violet region where the Higgs-mediated pair production dominates, and changed the discussion below Eq.~(4.9). The same mistake was done in Figure 8, which has also been replaced.
We apologize for these mistakes.

"In the case that the DM candidate is not stable, is there any advantage from X having a high spin? I believe, from simple dimensional analysis, that the lifetime will not depend on the spin, although it could be nice to explicitly comment on this point."

Indeed, the decay width is not significantly affected by the spin of the DM candidate. We added a comment about it below Eq.~(4.18).
We do not see clear advantages for going to higher spins in this scenario.
This is precisely the reason why we only provide some simple dimensional analysis for $S>2$, without going through a detailed analysis.
One thing one could mention here is that, because $c_H\sim M_{\rm Pl} m^{S-2}/\Lambda^{S-1}$, it is more natural for $S>2$ to arrive at small $c_H$ and satisfy the DM decay constraints.
We added a comment about it below Eq.~(4.16).

"Are the authors aware of explicit UV completions where the lightest stable particle is of high spin? Any idea why this could be the case?"

This is a very good question and indeed it is not so common to find examples of dark sectors where the lightest particle has a large spin. However, there is no theoretical obstruction for such type of spectrum and one can find some examples where this is realized. As is very well known, in supergravity it is possible to have the gravitino (spin-3/2) as the lightest stable particle in broad regions of the parameter space. Spin-2 DM can appear e.g. in bigravity.
For $S>2$, arguably the most promising direction is $SU(N)$ gauge theories in the limit of large $N$. In particular, 1707.05380 studied a model where the DM particle is a dark baryon with higher spin. In this case one can argue that the lightest dark baryon corresponds to the totally symmetric spin configuration, therefore $S = N/2$. Finally, in nuclear physics there is a plethora of higher-spin ground states, but that's in the context of a non-relativistic EFT with $\Lambda \approx m$, so it's probably not directly relevant from the point of view of our paper.

We added a comment about the possible large-$N$ $SU(N)$ origin of our DM particle in the 2nd paragraph of Introduction.

---

## Round 2 · Referee Report · Anonymous · 2021-1-25

Strengths

1 - Extensive analysis of higher spin DM freeze-in production though purely gravitational effects
2 - Discussion and analysis of the effects off Z2 breaking coupling on DM productions, mainly focussing on interactions with the Higgs sector of the Standard Model
3 - Extensive discussion (in text and appendix) of on-shell amplitude methods for the calculation of the amplitudes relevant for DM production.

Weaknesses

1 - The computational details and approximate equations, spread throughout the main text, make the paper difficult to follow. While it is great to have some semi analytic expression to understand the behavior of some physical process, some of those appearing in the paper do not provide any particular insight. I'm referring in particular to numerical expression like Eq. 3.18, 3.26-27, and formulas for the amplitudes like eq 3.3-3.5, 4.2, 4.5-4.7, ...

2 - The novelty of the paper is mainly in the computational details, related to the use of on-shell amplitudes.

3 - Given the complexity of some of the computations, a more detailed comparison with the existing literature [Criado et al. 2010.02224, Alexander et al. 2010.15125] (if possible) would be a great sanity check.

Report

The paper applies on-shell amplitudes techniques to study the phenomenology of higher spin DM produced by freeze-in.

The paper explore production through purely gravitational interactions and through Higgs mediated couplings.

The paper is well written and publication is recommended after some minor changes.

While the paper is very thorough in the description of the formalism, this can sometimes take the focus away from the actual phenomenology. A lot of formulas and semi analytical estimates are present in the main text but they do not necessarily improve understanding of the various physical process, on the other hand they tend to distract from the physics.

Some comparison with existing literature discussing higher spin DM production is welcome, especially as a sanity check of the various results.

Requested changes

- Is equation 3.7 as written valid for spins < 2 ?
- When this is possible, can the author compare their results with those of [Criado et al. 2010.02224] and [Alexander et al. 2010.15125] ?
- I would suggest moving some of the bigger helicity amplitude equations (3.3-3.5, 4.2, 4.5-4.7 in some appendix.

  • validity: good
  • significance: good
  • originality: good
  • clarity: ok
  • formatting: good
  • grammar: excellent

Author:  Giulia Isabella  on 2021-02-16  [id 1247]

(in reply to Report 2 on 2021-01-25)

We thank the referee for the helpful comments and all the interesting points raised. Below, we address the referee's points sequentially, and detail the changes we have made in the draft to address their helpful feedback. For convenience we list the questions and respective answers below.

"Is equation 3.7 as written valid for spins $< 2$ ?"

The production amplitudes in Eqs.~(3.3)-(3.5) are valid as they stand also for $S=3/2$, and therefore the scaling in Eq.~(3.7) remains valid for $S=3/2$. For $S=1$ Eqs.~(3.3)-(3.5) are also valid, but with the convention that the first term involving $\sum_{k=1}^{2S-2}$ should be set to zero. As this first term dominates the scaling for generic $S$, Eq.~(3.7) is not valid for $S=1$, and one instead has $M_{\rm ann} \sim E^2/M_{\mathrm {Pl}}^2$, as follows from the second term involving $\sum_{k=0}^{2S-2}$ in Eqs. (3.3)-(3.5). Eqs.~(3.3)-(3.5) are not valid for $S<1$.

We added a comment below Eq.~(3.7) that the scaling is valid for $S \ge 3/2$ and commented on the scaling for $S=1$. We also added footnote 6 commenting on the (in)validity of Eqs.~(3.3)-(3.5) for $S\leq 1$.

" When this is possible, can the author compare their results with those of [Criado et al. 2010.02224] and [Alexander et al. 2010.15125] ?"

We added the next-to-last paragraph in Conclusions where we elaborate on the similarities and differences between our work and those of [Criado et al. 2010.02224] and [Alexander et al. 2010.15125]. We elaborate more about the differences between the models in the footnote of pag.2: "The main difference between [35,36] and our work is that neither uses the massive on-shell formalism of [15]. Moreover, Ref. [36] consider the case of DM production DURING inflation and we are considering the production AFTER inflation. Regarding [35], the leading DM-Higgs interaction is a 4-point contact term, which for us is a 3-point DM-Higgs-Higgs coupling. "

"I would suggest moving some of the bigger helicity amplitude equations (3.3-3.5, 4.2, 4.5-4.7) in some appendix."

We agree with the referee that the formulas could distract from the phenomenological results. However, we believe that equations (3.3)-(3.5) and (4.5)-(4.7) (now 4.4-4.6) are well justified to be in the main text. First, to our knowledge, it is the first time that the amplitudes for gravitational scattering of massive spin-$S$ particles on SM matter appear in the literature. Thus, these equations represent an important original contribution of our paper, and they are one of our main results. Second, these formulas also show explicitly what we advocate in the introduction: the massive spinor formalism is well suitable to obtain compact formulas for scattering amplitudes of particles with general spins (the analogous formulas obtained in the standard Lagrangian formalism would be lengthy and opaque). Last but not least, it is from these formulas that we could see in a transparent way the energy scaling of the amplitudes, which were then used for the dimensional analysis estimations. So, the formulas indeed improve our understanding of the physical processes relevant for higher-spin DM production.

On the other hand, we agree that Eq.~(4.2) from the previous version is not essential for the subsequent discussion. Therefore we moved it to Appendix C, as Eq.~(C.1). In the case, of equation 4.2, since the theory is defined in terms of the 3-point amplitudes we quoted all possibilities. This coupling is not particularly relevant to the phenomenology but since it is the same as in the dRGT gravity, we found worthy to write it.

Other changes:

  • A coding mistake has been made when plotting the green region on Figure 6. The figure has been replaced, and now the green region matches the corresponding region in Figure 7. We also spotted a typo in the code related to the position of the violet region in Fig.~6 (Higgs-mediated pair production). After correcting this mistake, the violet region shifted to huge valus of $c_H$, which are not relevant withing the validity of the EFT. For this reason do not plot anymore the violet region where the Higgs-mediated pair production dominates, and changed the discussion below Eq.~(4.9). The same mistake was done in Figure 8, which has also been replaced. We apologize for these mistakes.
  • On page 2 we added footnote 1 specifying that the no-go theorem for massless higher spins holds for Poincare-invariant, unitary, local QFTs in four spacetime dimensions (a loophole exists e.g. for QFTs in AdS background).

---

## Round 3 · Referee Report · Anonymous (Referee 2) · 2021-4-14

Report

The new manuscript can be accepted for publication as is

---

## Round 3 · Referee Report · Anonymous (Referee 1) · 2021-4-23

Report

I would like to thank the authors for the reply to my previous report. With their clarifications and additions I believe the manuscript is now suitable for publication.

---

## Round 3 · Author Response

Dear Editor,
We thank the referees for the constructive comments and all the interesting points raised. Below, we detail the changes we have made in the draft to address their helpful feedback.

---

## Round 3 · List of Changes

- A mistake has been made by plotting the green region on Figure 6. The figure has been replaced, and now the green region corresponds to the same region in Figure 7. Thank to this comment we spotted a typo in the code related to the position
of the violet region (Higgs-mediated pair production). This region lies above $M_Pl$ and therefore is
relevant only beyond the validity of the EFT. For this reason we excluded it from the plot and changed
the discussion below Eq.(4.10). The same mistake was done in Figure 8, which has also been replaced.
- Comment added under equation 3.6 on the scaling of the Wilson coefficients
- Comment added below eq. (3.7) on the validity of eqs. 3.3-3.5
- We elaborate more about the di fferences between the models in the footnote of pag.2.

---

## Editorial Decision

published